# Unlabeled Compressive Sensing under Sparse Permutation and Prior Information

**Garweet Sresth**  *garweetsresth@gmail.com*
*Department of Electrical Engineering*
*IIT Bombay*

**Satish Mulleti**  *satish.mulleti@iitb.ac.in*
*Department of Electrical Engineering*
*IIT Bombay*

**Ajit Rajwade**  *ajitvr@cse.iitb.ac.in*
*Department of Computer Science and Engineering*
*IIT Bombay*

**Reviewed on OpenReview:** *https://openreview.net/forum?id=HaAg9RN7Hi*

## Abstract

In this paper, we study the problem of unlabeled compressed sensing, where the correspondence between the measurement values and the rows of the sensing matrix is lost, the number of measurements is less than the dimension of the regression vector, and the regression vector is sparse in the identity basis. Additionally, motivated by practical situations, we assume that we accurately know a small number of correspondences between the rows of the measurement matrix and the measurement vector. We propose a tractable estimator, based on a modified form of the LASSO, to estimate the regression vector, and we derive theoretical error bounds for the estimate. This is unlike previous approaches to unlabeled compressed sensing, which either do not produce theoretical bounds or which produce bounds for intractable estimators. We show that our algorithm outperforms a hard thresholding pursuit (HTP) approach and an $\ell_1$-norm estimator used to solve a similar problem across diverse regimes. We also propose a modified HTP based estimator which has superior properties to the baseline HTP estimator. Lastly, we show an application of unlabeled compressed sensing in image registration, demonstrating the utility of a few known point correspondences.

## 1 Introduction

Estimation of an unknown vector $\boldsymbol{\beta}^* \in \mathbb{R}^p$ from its (possibly noisy) linear measurements $\boldsymbol{A}\boldsymbol{\beta}^* \in \mathbb{R}^N$, where $\boldsymbol{A} \in \mathbb{R}^{N \times p}$ is the measurement matrix, is a well-studied problem that naturally arises in numerous fields. However, in some scenarios, the correspondences between the rows of the measurement matrix and the measurements get partially or entirely lost due to errors made during the measurement process. Specifically, the permuted and noisy measurements are given as

$$\boldsymbol{y} = \boldsymbol{P}\boldsymbol{A}\boldsymbol{\beta}^* + \boldsymbol{w}, \tag{1}$$

where $\boldsymbol{w} \in \mathbb{R}^N$ is the measurement noise vector and $\boldsymbol{P} \in \mathbb{R}^{N \times N}$ denotes an unknown permutation matrix. Hence, the problem reduces to that of estimating the unknown regression vector from an unknown permutation of its measurements. This problem, known as *unlabeled sensing*, appears in applications such as simultaneous localization and mapping in robotics (Thrun & Leonard, 2008), record linkage (Lahiri & Larsen, 2005), image or point cloud registration (Pan & Zhang, 2005; Li et al., 2021), security and privacy (Narayanan & Shmatikov, 2008), and simultaneous pose and correspondence determination in computer vision (Marques et al., 2009).

Unlabeled sensing is well studied theoretically in the over-determined regime where $N > p$ and typically, under the assumption that the entries of $\mathbf{A}$ are drawn i.i.d. from an arbitrary continuous probability distribution. Within these settings and without measurement noise, Unnikrishnan et al. (2018) and Han et al. (2018) state that every $\boldsymbol{\beta}^* \in \mathbb{R}^p$ can be uniquely recovered with probability one from its measurements $\boldsymbol{PA\beta}^*$ if $N \geq 2p$. Under some more assumptions on $\boldsymbol{\beta}^*$, the condition $N \geq 2p$ can be relaxed to $N \geq p+1$ (Tsakiris & Peng, 2019). The over-determined unlabeled sensing problem is challenging in the presence of noise. In Pananjady et al. (2018); Hsu et al. (2017), the authors have derived lower bounds on the signal-to-noise ratio (SNR) for approximate recovery of $\boldsymbol{\beta}^*$. Algorithms to solve the problem for different values of $p$ are suggested in Peng & Tsakiris (2020); Tsakiris & Peng (2019); Candes & Tao (2005); Slawski & Ben-David (2019). Sresth et al. (2024) extends the approach in Slawski & Ben-David (2019) by assuming that some correspondences between the rows of the measurement matrix and measurement vector are known.

An under-determined unlabeled sensing problem, where $N < p$, can be solved with some additional priors on $\boldsymbol{\beta^*}$. To the best of our knowledge, Peng et al. (2021) is the only work in the context of unlabeled compressive sensing. In Peng et al. (2021), the authors assumed that $\boldsymbol{\beta}^*$ is $k$-sparse (i.e., it has at most $k$ non-zero elements) and showed that $N \geq 2k$ measurements are sufficient for unique recovery in the absence of noise. Interestingly, the result is analogous to that proved for the over-determined unlabeled sensing problem from Unnikrishnan et al. (2018). On the algorithmic side, a hard-thresholding pursuit-based approach is proposed for unlabeled compressive sensing in Peng et al. (2021). However, no theoretical error bounds on the performance of their *algorithm* are established in Peng et al. (2021), and moreover, their algorithm involves a computationally expensive $\ell_1$-norm optimization step to determine the unknown vector $\boldsymbol{\beta^*}$. Also, Peng et al. (2021) employs a subgradient method in their algorithm where the function value may not always decrease across the iterations of the algorithm, as mentioned in their paper.

**Possible Prior Information:** In several unlabeled sensing applications, such as image alignment, record-linkage, etc., we encounter sparse permutation scenarios, that is, the correspondence between the measurements in $\boldsymbol{y}$ and the rows of $\boldsymbol{A}$ is incorrect for only a small fraction of measurements. Equivalently, we have that $\|\boldsymbol{PA\beta^*} - \boldsymbol{A\beta^*}\|_0 \ll N$. Moreover, in many applications, some of the correspondences between the rows of $\boldsymbol{A}$ and the rows of $\boldsymbol{y}$ are known a priori. For example, in the image alignment problem, many point correspondences in the two images being matched are obtained via feature point matching methods such as Lowe (2004), with permutations often occurring due to factors such as image self-similarity (i.e., similarity of patches in distant image regions). However, a domain expert can manually mark a few corresponding point pairs in the fixed and moving images, thus yielding some measurements with correspondences known in advance.

Nguyen & Tran (2012) propose a robust form of least absolute shrinkage and selection operator (Lasso), which can estimate the sparse regression vector from its grossly corrupted linear measurements. Treating the permuted measurements as gross corruptions, robust Lasso can be employed to solve unlabeled compressed sensing problems. However, as motivated previously, if there is a prior knowledge of some correspondences between the rows of $\boldsymbol{A}$ and the rows of $\boldsymbol{y}$, robust Lasso is not able to make use of it. A natural question is how the prior knowledge of some correspondences would improve the estimation of $\boldsymbol{\beta^*}$. Such questions are answered in an over-determined setup in Sresth et al. (2024). However, these questions are still open to under-determined, unlabeled sensing problems.

## 1.1 Our Contribution

Our paper presents the following contributions:

1. We propose two algorithms to solve the unlabeled compressed sensing problem which can make use of known correspondences is available:

   (i) Augmented Robust Lasso (Ar-Lasso), a modification of the standard least absolute shrinkage and selection operator (Lasso).

   (ii) Augmented Hard Thresholding Pursuit (A-Htp) which involves a gradient-descent step and computationally cheaper $\ell_2$-norm optimization step, rather than subgradient-method and $\ell_1$-norm optimization as required in the earlier approach proposed in Peng et al. (2021).

| Symbol | Meaning |
|---|---|
| $N$ | total number of measurements |
| $m$ | number of labeled measurements |
| $n := N - m$ | number of potentially unlabeled measurements |
| $p$ | signal dimension |
| $\boldsymbol{y} \in \mathbb{R}^N$ | vector of compressive measurements |
| $\boldsymbol{\beta^*} \in \mathbb{R}^p$ | unknown sparse signal vector |
| $\boldsymbol{A} \in \mathbb{R}^{N \times p}$ | sensing matrix |
| $\boldsymbol{e^*} \in \mathbb{R}^n$ | unknown sparse permutation corruption vector |
| $k$ | $\ell_0$ norm of $\boldsymbol{\beta^*}$ |
| $s$ | $\ell_0$ norm of $\boldsymbol{e^*}$ |
| $\boldsymbol{y}_1 \in \mathbb{R}^m$ | sub-vector of measurements with known correspondences |
| $\boldsymbol{y}_2 \in \mathbb{R}^n$ | sub-vector of measurements with potentially unknown correspondences |
| $\boldsymbol{w} \in \mathbb{R}^N$ | measurement noise vector |

Table 1: Glossary of symbols used in this paper.

AR-LASSO is suitable for both compressible and perfectly sparse regression vectors, unlike the earlier approach in Peng et al. (2021), which works only for perfectly sparse regression vectors. However, A-HTP can work only in the case of perfectly sparse regression vectors. Note that AR-LASSO and A-HTP algorithms are applicable even when there are no correspondences known. Moreover, apart from the unlabeled sensing problem, our algorithms are applicable in any regression problem where some measurements are grossly corrupted, and the measurements can be separated into two disjoint sets: (*i*) measurements without any gross corruption and (*ii*) measurements with possible gross corruption.

2. For both AR-LASSO and A-HTP algorithms, we derive theoretical upper bounds on the estimation error of the unknown vector in terms of the number of measurements $N$, number of permutations $s$, dimension of the unknown vector $p$, sparsity of the unknown vector $k$, number of known correspondences $m$, the number of measurements $n := N - m$ with potentially unknown correspondences, and measurement noise variance $\sigma^2$. Specifically, for AR-LASSO, we characterize the notion of a generalized, extended, restricted eigenvalue condition (GEREC), which enables us to prove performance guarantees for the algorithm. GEREC is a generalization of the extended, restricted eigenvalue condition studied in Nguyen & Tran (2012), to a scenario where some of the correspondences are (possibly) known in advance. We show that the family of Gaussian measurement matrices obeys GEREC with a high probability. We compare our error bounds to those obtained in Nguyen & Tran (2012) and demonstrate that the information of known correspondences allows us to tolerate a larger number of permutations and also results in a lower estimation error.

3. Further, we demonstrate a geometric convergence result for A-HTP under the condition that the sensing matrix obeys some form of restricted isometry property. A-HTP involves joint-estimation of $\boldsymbol{x}^*$, which is a row concatenation of $\boldsymbol{\beta^*}$ and the permutation corruption vector, through an augmented matrix $\left[ \begin{array}{c|c} \boldsymbol{A} & \begin{array}{c} \boldsymbol{0}_{m \times (N-m)} \\ \boldsymbol{I}_{(N-m) \times (N-m)} \end{array} \end{array} \right]$. In order to exploit the distribution of non-zero entries in $\boldsymbol{x}^*$, we introduce a notion of structured-sparsity restricted isometry property. Following this, we demonstrate that $N \geq C(k \log (p/3k) + s \log (n/3s))$ (where $C > 0$ is a constant) is a sufficient condition for an accurate recovery via A-HTP, which is a more relaxed requirement than $N \geq C((k+s) \log \frac{p+n}{3(k+s)})$ which is what one obtains from a naive analysis without exploiting the specific sparsity structure of $\boldsymbol{x}^*$.

4. Next, we compare our algorithms to the $\ell_1$-norm hard-thresholding pursuit approach from Peng et al. (2021) and to another $\ell_1$-norm-based estimator motivated from Candes & Tao (2005), across diverse regimes. We demonstrate that our algorithms outperform them across all the regimes examined.

5. Lastly, we demonstrate the impact of utilizing known correspondences in an image registration task. For this task, the problem of unlabeled compressed sensing with sparse permutations and a set of priors is especially relevant in the following manner: ($i$) In many image registration tasks, a domain expert can mark out a few point correspondences accurately. This provides prior information for the regression problem. ($ii$) In image registration, the underlying motion vector fields are sparse or compressible in universal dictionaries such as the discrete Fourier transform or discrete cosine transform (James et al., 2019). If salient feature point tracking is used for obtaining point correspondences, then the motion vectors at only a small set of points (in the image domain) are observed. This is therefore a compressed sensing or sparse recovery problem. ($iii$) A fraction of the computed point correspondence pairs suffer from permutation effects due to factors like self-similarity. This is therefore an unlabeled sensing problem where the permutation set is sparse. For more details on this, please refer to Section 1 and Figure 1 of the supplemental material.

6. Summarily, techniques presented in this paper are simple, but this is the first piece of work to present a theoretical analysis of unlabeled compressed sensing (a topic with *very* limited literature) for any *tractable* estimator. The key contributions in the proof techniques used include: (1) the proposition of generalized, extended, restricted eigenvalue condition (GEREC) and its use within the AR-LASSO bounds; (2) concept of structured-sparsity, restricted isometric property (SS-RIP) and the use of the specific form of structured sparsity in the $(p+n)$-dimensional vector $\boldsymbol{x}^* := \begin{bmatrix} \boldsymbol{\beta}^{*T} & \boldsymbol{e}^{*T} \end{bmatrix}^T$ being recovered – namely that the sub-vector corresponding to $\boldsymbol{\beta}^* \in \mathbb{R}^p$ contains $k$ non-zero elements, and the sub-vector corresponding to $\boldsymbol{e}^* \in \mathbb{R}^n$ contains $s$ non-zero elements. The use of such a structure to obtain a requirement on the minimum number of measurements for an accurate recovery of $\boldsymbol{x}^*$ via A-HTP is novel, and is not found in classic papers on model based compressive sensing such as Baraniuk et al. (2010).

## 2   Problem Formulation and Notations

Consider a set of linear measurements as in equation 1 with $N < p$ and the following assumptions:

(C1) $\boldsymbol{w} \sim \mathcal{N}(\boldsymbol{0}, \sigma^2 \boldsymbol{I}_N)$ and $\boldsymbol{A}^i \sim \mathcal{N}(\boldsymbol{0}, \boldsymbol{\Sigma})$ where $\boldsymbol{A}^i \in \mathbb{R}^p$ is the $i$-th row of the measurement matrix $\boldsymbol{A}$, and $\boldsymbol{I}_N$ stands for the $N \times N$ identity matrix. We denote the smallest eigenvalue of $\boldsymbol{\Sigma}$ by $C_{\min}(\boldsymbol{\Sigma})$, the largest eigenvalue of $\boldsymbol{\Sigma}$ by $C_{\max}(\boldsymbol{\Sigma})$ and the largest entry on the diagonal of $\boldsymbol{\Sigma}$ by $\xi(\boldsymbol{\Sigma})$.

(C2) The regression vector $\boldsymbol{\beta}^*$ is $k$-sparse where $k$ need not be known in advance. We denote the support of non-zero entries of $\boldsymbol{\beta}^*$ by the set $T$.

(C3) Any $m$ out of $N$ measurements have accurate correspondences where $m < p$. Without loss of generality, we assume that the correspondences of the top $m$ measurements are accurate, that is, $y_i = \boldsymbol{A}^i \boldsymbol{\beta}^* + w_i$ for $i = 1, 2, \ldots, m$.

(C4) In the remaining $n := N - m < p$ measurements, at most $s \ll n$ measurements have incorrect correspondences. However, we do not know the value of $s$ and which $s$ measurements have incorrect correspondences.

With assumptions (C1)-(C4), we decompose $\boldsymbol{y}$ in equation 1 as

$$\boldsymbol{y}_1 = \boldsymbol{A}_1 \boldsymbol{\beta}^* + \boldsymbol{w}_1 \text{ and } \boldsymbol{y}_2 = \boldsymbol{P}_2 \boldsymbol{A}_2 \boldsymbol{\beta}^* + \boldsymbol{w}_2, \tag{2}$$

where $\boldsymbol{y}_1 \in \mathbb{R}^m$ denotes the sub-vector of measurements with known correspondences, $\boldsymbol{y}_2 \in \mathbb{R}^n$ is the sub-vector of remaining measurements (with $m + n = N$) and $\boldsymbol{P}_2 \in \mathbb{R}^{n \times n}$ is an unknown permutation matrix. We denote $\boldsymbol{y} := \begin{bmatrix} \boldsymbol{y}_1^T & \boldsymbol{y}_2^T \end{bmatrix}^T$, $\boldsymbol{A} := \begin{bmatrix} \boldsymbol{A}_1^T & \boldsymbol{A}_2^T \end{bmatrix}^T$ and $\boldsymbol{w} := \begin{bmatrix} \boldsymbol{w}_1^T & \boldsymbol{w}_2^T \end{bmatrix}^T$. For further analysis, define the permutation corruption vector as

$$\boldsymbol{z}^* = \sqrt{n} \boldsymbol{e}^* := (\boldsymbol{P}_2 \boldsymbol{A}_2 \boldsymbol{\beta}^* - \boldsymbol{A}_2 \boldsymbol{\beta}^*) \in \mathbb{R}^n. \tag{3}$$

The assumption (C4) implies that $\boldsymbol{e}^*$ is $s$-sparse. We denote the support of non-zero entries of $\boldsymbol{e}^*$ by the set $S$. Using equation 3, we can write equation 2 as

$$\boldsymbol{y}_2 = \boldsymbol{A}_2\boldsymbol{\beta}^* + \sqrt{n}\boldsymbol{e}^* + \boldsymbol{w}_2. \tag{4}$$

The goal is to estimate $\boldsymbol{\beta}^*$ under assumptions (C1)-(C4). To this end, we provide the formulations for AR-LASSO and A-HTP algorithms in the next sections.

## 3 Augmented Robust LASSO

With the introduction of $\boldsymbol{e}^*$, the objective of estimating $\boldsymbol{\beta}^*$ from $\boldsymbol{y}$ under assumptions (C1)-(C4) is posed as a convex optimization problem. Specifically, we propose to minimize an augmented, robust, least-absolute shrinkage and selection operator (AR-LASSO)-based objective function, given by:

$$L(\boldsymbol{\beta},\boldsymbol{e}) := \frac{1}{2m}\|\boldsymbol{y}_1 - \boldsymbol{A}_1\boldsymbol{\beta}\|_2^2 + \frac{1}{2n}\|\boldsymbol{y}_2 - \boldsymbol{A}_2\boldsymbol{\beta} - \sqrt{n}\boldsymbol{e}\|_2^2 + \lambda_\beta\|\boldsymbol{\beta}\|_1 + \lambda_e\|\boldsymbol{e}\|_1. \tag{5}$$

Here, the first two terms are data-fit terms, and $\|\boldsymbol{\beta}\|_1$, $\|\boldsymbol{e}\|_1$ are the sparsity-promoting terms in the regression vector and corruption vector, respectively weighted by regularization parameters $\lambda_\beta > 0$ and $\lambda_e > 0$. Next, we formulate the optimization problem as

$$(\tilde{\boldsymbol{\beta}},\tilde{\boldsymbol{e}}) := \underset{\boldsymbol{\beta}\in\mathbb{R}^p,\boldsymbol{e}\in\mathbb{R}^n}{\arg\min}\ L(\boldsymbol{\beta},\boldsymbol{e}). \tag{6}$$

The question remains, under what conditions AR-LASSO is able to estimate $\boldsymbol{\beta}^*$ with a low error? In the next subsection, we discuss the generalized, extended, restricted eigenvalue condition based on which we present theoretical guarantees for AR-LASSO to answer such questions.

### 3.1 Theoretical Guarantees for AR-LASSO

First, define the following set:

$$\mathcal{C} := \{(\boldsymbol{h},\boldsymbol{f}) \in (\mathbb{R}^p \times \mathbb{R}^n)\ \text{such that}\ \|\boldsymbol{h}_{\boldsymbol{T}^C}\|_1 + \lambda\|\boldsymbol{f}_{\boldsymbol{S}^C}\|_1 \leq 3\|\boldsymbol{h}_{\boldsymbol{T}}\|_1 + 3\lambda\|\boldsymbol{f}_{\boldsymbol{S}}\|_1\}, \tag{7}$$

where $\lambda = \frac{\lambda_e}{\lambda_\beta}$. It can be shown that the error term $(\tilde{\boldsymbol{\beta}} - \boldsymbol{\beta}^*, \tilde{\boldsymbol{e}} - \boldsymbol{e}^*)$ belongs to the set $\mathcal{C}$ (see the proof of Theorem 1 in the underline{supplemental material}, especially the part from equations (S3) to (S13)). The set $\mathcal{C}$ is a natural restriction on the error vectors $\boldsymbol{h} := \tilde{\boldsymbol{\beta}} - \boldsymbol{\beta}^*$ and $\boldsymbol{f} := \tilde{\boldsymbol{e}} - \boldsymbol{e}^*$ that emerges from the optimization problem in equation 6. Hence expectedly, the theoretical guarantees for AR-LASSO rely on the following two key properties of sensing matrix $\boldsymbol{A}$ over the restricted set $\mathcal{C}$:

(1) **Generalized, extended, restricted eigenvalue condition** (GEREC): We say that the measurement matrix $\boldsymbol{A}$ obeys the generalized, extended, restricted eigenvalue condition on the restricted set $\mathcal{C}$ if there exists $k_e > 0$ such that the following holds:

$$k_e(\|\boldsymbol{h}\|_2^2 + \|\boldsymbol{f}\|_2^2) \leq \frac{1}{N}\|\boldsymbol{A}\boldsymbol{h}\|_2^2 + \frac{n}{N}\|\boldsymbol{f}\|_2^2, \ \forall\ (\boldsymbol{h},\boldsymbol{f}) \in \mathcal{C}. \tag{8}$$

The GEREC is essentially stating that the sum of the squared magnitudes of the vectors $\boldsymbol{A}\boldsymbol{h}$ and $\boldsymbol{f}$ is lower-bounded by a constant factor times $\|\boldsymbol{h}\|_2^2 + \|\boldsymbol{f}\|_2^2$ for any $(\boldsymbol{h},\boldsymbol{f}) \in \mathcal{C}$. Recall that $n \leq N$. The above condition is a generalization of the extended, restricted eigenvalue condition studied in Nguyen & Tran (2012), to the scenario where some of the correspondences are known, that is when $n \leq N$. To give more context, we use the above condition with $(\boldsymbol{h},\boldsymbol{f}) := (\tilde{\boldsymbol{\beta}} - \boldsymbol{\beta}^*, \tilde{\boldsymbol{e}} - \boldsymbol{e}^*)$ for theoretical analysis of AR-LASSO. Essentially, $\boldsymbol{f}$ plays the role of error in estimation of the permutation corruption vector $\boldsymbol{z}^* := \boldsymbol{P}_2\boldsymbol{A}_2\boldsymbol{\beta}^* - \boldsymbol{A}_2\boldsymbol{\beta}^*$. Note the difference between GEREC and the extended, restricted eigenvalue condition in Nguyen & Tran (2012). In Nguyen & Tran (2012), the authors assume that each of the $N$ measurements can undergo gross corruption and the aim is to lower-bound the term $(\frac{1}{N}\|\boldsymbol{A}\boldsymbol{h}\|_2^2 + \|\boldsymbol{f}\|_2^2)$ where $\boldsymbol{A} \in \mathbb{R}^{N\times p}, \boldsymbol{h} \in \mathbb{R}^p$ and $\boldsymbol{f} \in \mathbb{R}^N$ (see comment 3(a) after Lemma 1 in Nguyen & Tran (2012)). On

the other hand, we have assumed knowledge of $m$ correspondences, and hence, we are only concerned with permutation errors in the remaining $n = N - m$ measurements. Due to this, we have $\boldsymbol{f} \in \mathbb{R}^n$ as opposed to $\boldsymbol{f} \in \mathbb{R}^N$ as in Nguyen & Tran (2012). Alternatively, our $\boldsymbol{f}$ can be thought of as an $N$-dimensional vector whose top $m$ entries are zero. When none of the correspondences are known, that is, when $m = 0, n = N$, the generalized, extended, restricted eigenvalue condition reduces to the extended, restricted eigenvalue condition defined in Nguyen & Tran (2012).

(2) **Mutual incoherence condition** (Mic) (Nguyen & Tran, 2012): We require that there exists $k_m > 0$ such that

$$\frac{1}{\sqrt{n}}|\langle \boldsymbol{A_2 h}, \boldsymbol{f}\rangle| \le k_m(\|\boldsymbol{h}\|_2 + \|\boldsymbol{f}\|_2)^2 \ \forall \ (\boldsymbol{h}, \boldsymbol{f}) \in \mathcal{C}. \tag{9}$$

We refer to $k_m > 0$ as the mutual incoherence constant. The mutual incoherence condition allows decoupling the two error terms $\|\tilde{\boldsymbol{\beta}} - \boldsymbol{\beta^*}\|_2$ and $\|\tilde{\boldsymbol{e}} - \boldsymbol{e^*}\|_2$ given that we use $(\boldsymbol{h}, \boldsymbol{f}) := (\tilde{\boldsymbol{\beta}} - \boldsymbol{\beta^*}, \tilde{\boldsymbol{e}} - \boldsymbol{e^*})$ for theoretical analysis of Ar-Lasso. Also, the mutual coherence condition states that the column vectors of $\boldsymbol{A}$ are sufficiently dissimilar compared to columns of the identity matrix. This ensures that there is clear separation between the signal vectors $\boldsymbol{\beta^*}$ and $\boldsymbol{e^*}$ during compressive recovery.

Next, the question arises: Under what conditions does a measurement matrix obey Gerec and Mic? It can be shown that the family of Gaussian sensing matrices satisfies Gerec with suitable assumptions on $(N, m, s, k)$. To this end, we state the following lemma (proof in supplemental material):

**Lemma 1** (**Generalized, extended, restricted eigenvalue condition**). *Consider the Gaussian sensing matrix $\boldsymbol{A} \in \mathbb{R}^{N \times p}$ whose rows are i.i.d. $\mathcal{N}(\boldsymbol{0}, \boldsymbol{\Sigma})$. We have the set $\mathcal{C} := \{(\boldsymbol{h}, \boldsymbol{f}) \in (\mathbb{R}^p \times \mathbb{R}^n)$ such that $\|\boldsymbol{h_{T^c}}\|_1 + \lambda \|\boldsymbol{f_{S^c}}\|_1 \le 3\|\boldsymbol{h_T}\|_1 + 3\lambda\|\boldsymbol{f_S}\|_1\}$ with $|T| = k$ and $|S| = s$ as defined earlier. Select $\lambda = \rho\sqrt{\frac{n}{N}\frac{\log n}{\xi(\boldsymbol{\Sigma})\log p}}$, where $\rho \in (0, 1)$ is a constant. If $s \le c_1\frac{N}{\rho^2 \log n}$ and $N \ge c_2\frac{\xi(\boldsymbol{\Sigma})}{C_{min}(\boldsymbol{\Sigma})}k\log p$, then the following inequality holds with probability at least $1 - c_3\exp(-c_4 N)$: $min\left(\frac{C_{min}(\boldsymbol{\Sigma})}{128}, \frac{n}{8N}\right)(\|\boldsymbol{h}\|_2^2 + \|\boldsymbol{f}\|_2^2) \le \frac{1}{N}\|\boldsymbol{Ah}\|_2^2 + \frac{n}{N}\|\boldsymbol{f}\|_2^2 \ \forall \ (\boldsymbol{h}, \boldsymbol{f}) \in \mathcal{C}$, where $c_1, c_2, c_3, c_4$ are positive constants.*

We make the following observations: ($i$) We reiterate that the knowledge of $m$ correspondences requires us to only consider $\boldsymbol{f} \in \mathbb{R}^n$ as opposed to considering $\boldsymbol{f} \in \mathbb{R}^N$. As a result, the sensing matrix $\boldsymbol{A}$ obeys Gerec with a larger tolerance on $s$, that is, $s \le c_1\frac{N}{\rho^2 \log n} = c_1\frac{N}{\rho^2 \log(N-m)}$. As $m$ increases, a larger number of permutations can be tolerated. ($ii$) In the absence of known correspondences, that is, when $n = N$, the requirement on $s$ becomes $s \le c_1\frac{N}{\rho^2 \log N}$ which is consistent with that obtained in Lemma 1 of Nguyen & Tran (2012). ($iii$) The choice of $\lambda$ follows from Theorem 1 in the next section.

Moreover, Nguyen & Tran (2012) prove that the family of Gaussian sensing matrices also satisfies the mutual incoherence condition, again with some suitable assumptions on $(n, s, k)$. For the sake of completeness, we state the following lemma:

**Lemma 2** (**Mutual incoherence condition (Nguyen & Tran, 2012)**). *Consider the Gaussian sensing matrix $\boldsymbol{A_2} \in \mathbb{R}^{n \times p}$ whose rows are i.i.d. $\mathcal{N}(\boldsymbol{0}, \boldsymbol{\Sigma})$. We have the set $\mathcal{C} = \{(\boldsymbol{h}, \boldsymbol{f}) \in (\mathbb{R}^p \times \mathbb{R}^n)$ such that $\|\boldsymbol{h_{T^c}}\|_1 + \lambda \|\boldsymbol{f_{S^c}}\|_1 \le 3\|\boldsymbol{h_T}\|_1 + 3\lambda\|\boldsymbol{f_S}\|_1\}$ with $|T| = k$ and $|S| = s$ as defined earlier. Select $\lambda = \rho\sqrt{\frac{n}{N}\frac{\log n}{\xi(\boldsymbol{\Sigma})\log p}}$, where $\rho \in (0, 1)$ is a constant. Assume that $s \le min\left(\frac{N}{n}\frac{\xi(\boldsymbol{\Sigma})}{C_{min}(\boldsymbol{\Sigma})}\frac{k\log p}{\rho^2 \log n}, c_5\frac{C_{min}(\boldsymbol{\Sigma})}{C_{max}(\boldsymbol{\Sigma})}n\right)$ and $n \ge c_6\xi(\boldsymbol{\Sigma})\frac{C_{max}(\boldsymbol{\Sigma})}{C_{min}^2(\boldsymbol{\Sigma})}k\log p$ for some sufficiently small positive constant $c_5$ and sufficiently large constant $c_6$, then the following inequality holds with probability greater than $1 - \exp(-c_7 n)$: $\frac{1}{\sqrt{n}}|\langle \boldsymbol{A_2 h}, \boldsymbol{f}\rangle| \le k_m(\|\boldsymbol{h}\|_2 + \|\boldsymbol{f}\|_2)^2 \ \forall \ (\boldsymbol{h}, \boldsymbol{f}) \in \mathcal{C}$, where $c_7, k_m$ are positive constants. We refer to $k_m$ as the mutual incoherence constant.*

We pick the above lemma from Nguyen & Tran (2012) with a slight change in the expression for $\lambda$ to suit our problem. Note that the expression for $\lambda$ naturally follows from the theoretical analysis of Ar-Lasso (see Theorem 1). Consequently, to obey the mutual incoherence condition, the requirement on $s$ gets relaxed, that is, $s \le \frac{N}{n}\frac{\xi(\boldsymbol{\Sigma})}{C_{\min}(\boldsymbol{\Sigma})}\frac{k\log p}{\rho^2 \log n} = \frac{N}{N-m}\frac{\xi(\boldsymbol{\Sigma})}{C_{\min}(\boldsymbol{\Sigma})}\frac{k\log p}{\rho^2 \log n}$. As $m$ increases, more permutations can be tolerated.

For a consistency check, setting $m = 0$ gives us the same upper limit on $s$ as obtained in Lemma 2 in Nguyen & Tran (2012).

Now, since we have established that the family of Gaussian sensing matrices obey GEREC and MIC with a high probability, we move on to state the theoretical guarantees for AR-LASSO. Given the optimization problem equation 6, we present an upper bound on the estimation error $\|\tilde{\boldsymbol{\beta}} - \boldsymbol{\beta^*}\|_2 + \|\tilde{\boldsymbol{e}} - \boldsymbol{e^*}\|_2$ in the following theorem:

**Theorem 1.** *Consider the optimization problem in equation 6 and the observation model in equation 1 under the assumptions (C1)-(C4). Select $\lambda_\beta = \frac{2}{\rho} \frac{\|\boldsymbol{A^T w}\|_\infty}{N}$ and $\lambda_e = \frac{2\sqrt{n}}{N} \|\boldsymbol{w_2}\|_\infty$. Assume that $s \leq min\left( c_1 \frac{N}{\rho^2 \log n}, \frac{N}{n} \frac{\xi(\boldsymbol{\Sigma})}{C_{min}(\boldsymbol{\Sigma})} \frac{k \log p}{\rho^2 \log n}, c_5 \frac{C_{min}(\boldsymbol{\Sigma})}{C_{max}(\boldsymbol{\Sigma})} n \right)$ and that $N \geq c_2 \frac{\xi(\boldsymbol{\Sigma})}{C_{min}(\boldsymbol{\Sigma})} k \log p$, $n \geq c_6 \xi(\boldsymbol{\Sigma}) \frac{C_{max}(\boldsymbol{\Sigma})}{C_{min}^2(\boldsymbol{\Sigma})} k \log p$, so that $\boldsymbol{A}_2$ satisfies MIC with sufficiently small mutual incoherence constant $k_m$ such that $k_m < min\left( \frac{N}{n} \frac{C_{min}(\boldsymbol{\Sigma})}{512}, \frac{1}{32} \right)$, and $\boldsymbol{A}$ satisfies GEREC with constant $k_e > 0$. Then there exists positive constant $k_l > 0$ such that the following inequality holds with probability greater than $1 - 2/p - 2/n$:*

$$\|\tilde{\boldsymbol{\beta}} - \boldsymbol{\beta^*}\|_2 + \|\tilde{\boldsymbol{e}} - \boldsymbol{e^*}\|_2 \leq 6\sigma k_l^{-2} \max\left( \frac{1}{\rho} \sqrt{\frac{\xi(\boldsymbol{\Sigma}) k \log p}{N}}, \sqrt{\frac{n}{N}} \frac{s \log n}{N} \right). \tag{10}$$

A few insights on the theorem are in order:

1. In the noiseless scenario ($\sigma = 0$), the error term turns out to be zero. And hence, exact recovery of $(\boldsymbol{\beta^*}, \boldsymbol{e^*})$ is possible with a high probability. In the case of noisy measurements, note that the estimation error scales linearly with the measurement noise standard deviation $\sigma$.

2. For fixed $p$, $m$ and $n$, the error term scales as $\max(\sqrt{k}, \sqrt{s})$, i.e. sparser $\boldsymbol{\beta^*}$ and $\boldsymbol{e^*}$ yield more accurate reconstructions.

3. As pointed out earlier, the $m$ known correspondences allow $\boldsymbol{A}$ to obey GEREC and MIC with a higher tolerance on $s$. These are sufficient conditions for error bounds for AR-LASSO in Theorem 1 to hold. Moreover, note that the term $\sqrt{\frac{n}{N}} \frac{s \log n}{N}$ in equation 10 decreases as more correspondences are known. The higher the value of $m$ is for a fixed $n$ or a fixed $N$, the lower is the estimation error.

4. $\rho$ is a positive constant that controls the sparsity level in the regression vector and sparse error vector. If there are a large number of permutations expected, then a larger value of $\rho$ should be used for specifying $\lambda_\beta$ and vice-versa.

5. The choice of $\lambda$ in Lemma 1 and Lemma 2 follows from the expressions for $\lambda_\beta$ and $\lambda_e$ in Theorem 1. From the Gaussian concentration results, we know that $\|\boldsymbol{A^T w}\|_\infty \leq 2\sigma \sqrt{\xi(\boldsymbol{\Sigma}) N \log p}$ with probability at least $1 - 2/p$ and $\|\boldsymbol{w_2}\|_\infty \leq 2\sigma \sqrt{\log n}$ with probability at least $1 - 2/n$. Plugging the expressions in $\lambda = \lambda_\beta / \lambda_e$ gives us $\lambda = \rho \sqrt{\frac{n}{N} \frac{\log n}{\xi(\boldsymbol{\Sigma}) \log p}}$. On the other hand, Nguyen & Tran (2012) selects $\lambda = \rho \sqrt{\frac{\log N}{\xi(\boldsymbol{\Sigma}) \log p}}$ which is same when we put $m = 0$ in our expression for $\lambda$.

6. When no correspondences are known in advance, that is, $m = 0$ and $n = N$, we get the same error bound as obtained in Corollary 1 in Nguyen & Tran (2012).

7. Our approach casts unlabeled sensing with sparse permutations in a structured sparsity framework. This theorem presents performance bounds for it. The particular form of structured sparsity considered here is different from forms such as tree-structured and block-structured sparsity as analyzed in previous works like Baraniuk et al. (2010).

8. If $m$ is large enough so that the matrix $\boldsymbol{A_2}$ obeys the restricted eigenvalue condition, then we may not even require the permuted measurements, i.e. we can consider $N = m$. In such a case, the problem just reduces to a simple LASSO and the error bound would be as follows:

$$\|\tilde{\boldsymbol{\beta}} - \boldsymbol{\beta^*}\|_2 \leq O(\sigma \sqrt{k \log p/m}). \tag{11}$$

---

**Algorithm 1** Augmented Hard-Thresholding Pursuit

---

**Input**: Measurement vector $\boldsymbol{y}$, augmented matrix $\boldsymbol{H}$, sparsity level $k$ and number of permutations $s$ (both $k$ and $s$ can be estimated via cross-validation – see Sec. 5 under 'Choice of parameters')
**Parameter**: Learning rate $\mu$
**Output**: Estimate of $\boldsymbol{\beta}^*$

  1: $\boldsymbol{x}^{(0)} = \boldsymbol{0}, t = 0$.
  2: **while** not converged **do**
  3:     $\boldsymbol{l}^{(t+1)} = \boldsymbol{x}^{(t)} - \mu \boldsymbol{H}^T(\boldsymbol{H}\boldsymbol{x}^{(t)} - \boldsymbol{y})$
  4:     $S^{(t+1)} = \{\text{indices of } k \text{ largest entries of } \boldsymbol{l}^{(t+1)}(1 : p)\} \cup \{\text{indices of } s \text{ largest entries of } \boldsymbol{l}^{(t+1)}(p + 1 : p + n)\}$
  5:     $\boldsymbol{x}^{(t+1)} = \underset{\substack{\boldsymbol{x} \in \mathbb{R}^{p+n} \text{ s.t.} \\ \text{support}(\boldsymbol{x}) \subseteq S^{(t+1)}}}{\arg\min} \|\boldsymbol{H}\boldsymbol{x} - \boldsymbol{y}\|_2$
  6:     $t = t + 1$
  7: **end while**
  8: **return** $\boldsymbol{x}^{(t)}(1 : p)$

---

Sensing matrices randomly generated by Rademacher or Gaussian distributions would need to contain at least $O(k \log p)$ rows in order to recover $k$-sparse vectors in $\mathbb{R}^p$ successfully. Then the error bounds in Theorem 1 will reduce to the LASSO bounds in equation 11 since $N = m$. However such a large $m$ will generally not be available in applications. If $m$ is smaller than $O(k \log p)$, then successful recovery will not be achievable, and hence using the sparsely permuted measurements will be inevitable.

## 4 Augmented Hard-Thresholding Pursuit

Note that under assumptions (C1)-(C4), $\boldsymbol{y}$ can be re-written as

$$\boldsymbol{y} = \boldsymbol{H}\boldsymbol{x}^* + \boldsymbol{w}, \text{ where } \boldsymbol{H} := \begin{bmatrix} \boldsymbol{A}_1 & \boldsymbol{0}_{m \times n} \\ \boldsymbol{A}_2 & \boldsymbol{I}_{n \times n} \end{bmatrix}$$

is the augmented matrix and $\boldsymbol{x}^* := \begin{bmatrix} \boldsymbol{\beta}^{*T} & \boldsymbol{z}^{*T} \end{bmatrix}^T$. Recalling the notation $\boldsymbol{y} := \begin{bmatrix} \boldsymbol{y}_1^T & \boldsymbol{y}_2^T \end{bmatrix}^T$ and $\boldsymbol{w} := \begin{bmatrix} \boldsymbol{w}_1^T & \boldsymbol{w}_2^T \end{bmatrix}^T$, the problem of estimation of $\boldsymbol{x}^*$ can be posed as

$$\underset{\substack{\boldsymbol{x} = [\boldsymbol{\beta}^T \ \boldsymbol{z}^T]^T \\ \boldsymbol{\beta} \in \mathbb{R}^p, \boldsymbol{z} \in \mathbb{R}^n}}{\arg\min} \|\boldsymbol{y} - \boldsymbol{H}\boldsymbol{x}\|_2 \text{ s.t. } \|\boldsymbol{\beta}\|_0 \leq k, \|\boldsymbol{z}\|_0 \leq s. \tag{12}$$

We solve equation 12 using a modified form of the *hard thresholding pursuit* approach Foucart (2011) to suit the specific sparsity structure of our problem. The complete procedure is presented in Alg. 1. We refer to this approach as Augmented Hard Thresholding Pursuit or A-HTP, as it involves sparse recovery of an unknown vector (that is $\boldsymbol{x}^*$ here) using the *augmented* matrix $\boldsymbol{H}$. In Alg. 1, we note the following: (*i*) Line 3 is a standard gradient descent step where $\mu$ is the specified learning rate; (*ii*) Line 4 determines the support of $\boldsymbol{\beta}$ by selecting the indices corresponding to the $k$ largest absolute-value entries in $\boldsymbol{x}(1 : p)$, and the support of $\boldsymbol{e}$ by selecting the indices corresponding to the $s$ largest absolute-value entries in $\boldsymbol{x}(p+1 : p+n)$. The two support sets are combined to obtain the support set of $\boldsymbol{x}$; (*iii*) Line 5 is the debiasing step, and it re-estimates $\boldsymbol{x}$ over the support obtained in the previous step. This step involves computing the Moore-Penrose pseudo inverse of a submatrix of $\boldsymbol{H}$ containing all and only those columns appearing at indices in $S^{(t+1)}$. This step is computationally less expensive than the $\ell_1$-norm optimization step required in Peng et al. (2021). Moreover, Peng et al. (2021) employs the subgradient method in their algorithm where the function value may even increase, whereas our A-HTP is a gradient-descent-based method. In the next subsection, we provide a theoretical convergence analysis for A-HTP.

### 4.1 Convergence Analysis for A-HTP

In the compressed sensing framework, a condition for an accurate recovery of the unknown vector is typically based on the restricted isometry property (RIP) of the measurement matrix (Candes & Tao, 2005). A-HTP is inherently a compressed sensing algorithm. In this subsection, we state a theoretical guarantee on the recovery of $\boldsymbol{x}^*$ via A-HTP using some form of the restricted isometry constant (RIC) of the augmented measurement matrix $\boldsymbol{H}$. First, we recall the definition of RIC of a matrix. A matrix $\boldsymbol{A} \in \mathbb{R}^{N \times p}$ is said to satisfy the RIP of order $t$ if there exists a constant $\delta_t \in (0, 1)$ such that

$$(1 - \delta_t)\|\boldsymbol{y}\|_2^2 \leq \|\boldsymbol{A}\boldsymbol{y}\|_2^2 \leq (1 + \delta_t)\|\boldsymbol{y}\|_2^2 \tag{13}$$

holds for every $t$-sparse $\boldsymbol{y} \in \mathbb{R}^p$. The smallest constant $\delta_t$ for which this holds is called the order-$t$ RIC of the matrix $\boldsymbol{A}$. In other words, the application of $\boldsymbol{A}$ on any $t$-sparse vector approximately preserves the vector's norm.

Interestingly, in our problem, there is some structure in the distribution of non-zero entries in $\boldsymbol{x}^*$, that is, $\|\boldsymbol{x}^*(1:p)\|_0 \leq k$ and $\|\boldsymbol{x}^*(p+1:p+n)\|_0 \leq s$. To exploit this structured sparsity, we characterize a notion of structured-sparsity, restricted isometric property (SS-RIP) in the following definition.

**Definition 1.** *We say that the matrix $\boldsymbol{H}$ obeys the structured-sparsity, restricted isometric property (SS-RIP) of order $[(p,k),(n,s)]$ if there exists a restricted isometry constant $\delta := \delta_{[(p,k),(n,s)]}^{SS} \in (0,1)$ such that*

$$(1 - \delta)\|\boldsymbol{g}\|_2^2 \leq \|\boldsymbol{A}\boldsymbol{g}\|_2^2 \leq (1 + \delta)\|\boldsymbol{g}\|_2^2 \tag{14}$$

*holds for every $\boldsymbol{g} \in \mathbb{R}^{p+n}$ with $\|\boldsymbol{g}(1:p)\|_0 \leq k$ and $\|\boldsymbol{g}(p+1:p+n)\|_0 \leq s$.*

With this, we state a geometric convergence result on A-HTP:

**Theorem 2.** *Consider the Gaussian sensing matrix $\boldsymbol{A}$ whose entries are i.i.d. $\mathcal{N}(0, 1/N)$. Suppose that $\delta_{[(p,3k),(n,3s)]}^{SS}$ of the augmented measurement matrix $\boldsymbol{H}$ satisfies $\delta_{[(p,3k),(n,3s)]}^{SS} \leq \frac{1}{\sqrt{3}}$. Then the sequence $\boldsymbol{x}^{(t)}$ defined by A-HTP satisfies the following inequality $\forall\, t \geq 0$:*

$$\|\boldsymbol{x}^{(t)} - \boldsymbol{x}^*\|_2 \leq \gamma^t \|\boldsymbol{x}^{(0)} - \boldsymbol{x}^*\|_2 + \tau \frac{1 - \gamma^t}{1 - \gamma}\|\boldsymbol{w}\|_2, \tag{15}$$

*where $\tau := \dfrac{\sqrt{2\left(1 - \delta_{[(p,2k),(n,2s)]}^{SS}\right)} + \sqrt{1 + \delta_{[(p,k),(n,s)]}^{SS}}}{1 - \delta_{[(p,2k),(n,2s)]}^{SS}}$, $\gamma := \sqrt{\dfrac{2\delta_{[(p,3k),(n,3s)]}^{SS}{}^2}{1 - \delta_{[(p,2k),(n,2s)]}^{SS}{}^2}} < 1$ and $\boldsymbol{w}$ denotes the measurement noise vector from equation 1.*

The proof of the above theorem follows along the lines of the work in Foucart (2011, Theorem 3.8). The following lemma (proved in the supplemental material) outlines under what scenarios the augmented matrix $\boldsymbol{H}$ satisfies the criterion $\delta_{[(p,3k),(n,3s)]}^{SS} \leq \frac{1}{\sqrt{3}}$.

**Lemma 3.** *Consider the Gaussian sensing matrices $\boldsymbol{A_1} \in \mathbb{R}^{m \times p}$ and $\boldsymbol{A_2} \in \mathbb{R}^{n \times p}$ with i.i.d. $\mathcal{N}(0, 1/N)$ entries. There exist positive constants $c_8, c_9$ such that the augmented matrix $\boldsymbol{H} := \begin{bmatrix} \boldsymbol{A_1} & \boldsymbol{0}_{m \times n} \\ \boldsymbol{A_2} & \boldsymbol{I}_{n \times n} \end{bmatrix}$ satisfies the structured-sparsity restricted isometry property (SS-RIP) of order $[(p,k),(n,s)]$ provided that $k \log(p/k) + s \log(n/s) \leq c_8 N$, with probability at least $1 - 3 \exp(-c_9 N)$. The constants $c_8, c_9$ depend on the RIC $\delta$. Equivalently, we have the following result for all $\boldsymbol{x} \in \mathbb{R}^{p+n}$ with $\|\boldsymbol{x}(1:p)\|_0 \leq k$ and $\|\boldsymbol{x}(p+1:p+n)\|_0 \leq s$:*

$$\mathbb{P}\left((1 - \delta)\|\boldsymbol{x}\|_2^2 \leq \|\boldsymbol{H}\boldsymbol{x}\|_2^2 \leq (1 + \delta)\|\boldsymbol{x}\|_2^2\right) \geq 1 - 3 \exp(-c_9 N). \tag{16}$$

**Remarks on Theorem 2 and Lemma 3:**

1. As expected, the error bound (specifically, the second term) in Theorem 2 increases with the noise standard deviation. As $\gamma < 1$, the first term of the bound decreases with the number of iterations, i.e. $t$.

2. From Lemma 3, we require that $N \geq C(k \log (p/3k) + s \log (n/3s))$ for $\boldsymbol{H}$ to obey SS-RIP of order $[(p, 3k), (n, 3s)]$. And hence, from Theorem 2, it follows that $N \geq C(k \log (p/3k) + s \log (n/3s))$ is sufficient for an accurate recovery of $\boldsymbol{x}^*$ via A-HTP.

3. Since $n = N - m$, the requirement $N \geq C(k \log (p/3k) + s \log (n/3s))$ relaxes as more correspondences are known.

4. In the scenario $n = 0$, that is, when all the correspondences are known, the problem simplifies to the standard compressed sensing problem. Consequently, we require $N \geq C(k \log (p/3k))$ measurements for an accurate recovery of $\boldsymbol{\beta}^*$, which is consistent with the well-known requirement for compressed sensing via HTP.

**Comparison with the standard RIP condition.** Note that if we do not exploit the specific sparsity structure in $\boldsymbol{x}^*$, a condition on an accurate recovery of $\boldsymbol{x^*}$ via A-HTP is that the matrix $\boldsymbol{H}$ should satisfy RIP of order $3(k + s)$ Foucart (2011), which requires $N \geq C(k + s) \log (p + n)/(3(k + s))$. On the other hand, by exploiting the sparsity structure in $\boldsymbol{x}^*$, we demonstrate that $N \geq C(k \log (p/3k) + s \log (n/3s))$ is sufficient for an accurate recovery of $\boldsymbol{x}^*$ via our A-HTP algorithm. This is less than $C(k + s) \log [(p + n)/(3(k + s))]$.

## 5 Numerical Experiments

In order to assess the impact made by knowledge of known correspondences, we compare AR-LASSO from equation 6 and A-HTP from Alg. 1 to the following estimators, none of which use the prior information of known correspondences.:

($i$) The robust LASSO (R-LASSO) estimator given by $\underset{\boldsymbol{\beta} \in \mathbb{R}^p, \boldsymbol{e} \in \mathbb{R}^n}{\arg \min} \frac{1}{2N} \|\boldsymbol{y} - \boldsymbol{A}\boldsymbol{\beta} - \sqrt{n}\boldsymbol{e}\|_2^2 + \lambda_\beta \|\boldsymbol{\beta}\|_1 + \lambda_e \|\boldsymbol{e}\|_1$, which is effectively AR-LASSO with $m = 0$ and $N = n$.

($ii$) The $\ell_1$-norm hard thresholding pursuit approach in Peng et al. (2021) that minimizes $\|\boldsymbol{y} - \boldsymbol{A}\boldsymbol{\beta}\|_1$ w.r.t. $\boldsymbol{\beta} \in \mathbb{R}^p$ such that $\|\boldsymbol{\beta}\|_0 \leq k$. We refer to this approach as $\ell_1$-HTP.

($iii$) The $\ell_1 - \ell_1$ estimator motivated from Candes & Tao (2005); Candes et al. (2005) that is posed as $\underset{\boldsymbol{\beta} \in \mathbb{R}^p}{\arg \min} \|\boldsymbol{y} - \boldsymbol{A}\boldsymbol{\beta}\|_1 + \lambda_\beta \|\boldsymbol{\beta}\|_1$.

($iv$) The sparse Bayesian learning approach (SBL) approach Wipf & Rao (2004), which is an expectation-maximization (EM) based Bayesian method for compressed sensing. This algorithm has been shown to outperform several other non-learning based compressed sensing algorithms in Marques et al. (2018), and hence we use it in this paper to estimate both $\boldsymbol{\beta}^*$ and $\boldsymbol{e^*}$. Note that this approach has not been used in the literature in the context of unlabeled sensing, to our best knowledge. In this method, given knowledge of Gaussian noise in $\boldsymbol{y}$, we model the following probabilities, taking into account the definition of $\boldsymbol{H}$ and $\boldsymbol{x^*}$ from Sec. 4:

$$p(\boldsymbol{y}|\boldsymbol{x^*}) = \frac{\exp\left(-\|\boldsymbol{y} - \boldsymbol{H}\boldsymbol{x^*}\|_2^2/(2\sigma^2)\right)}{\sigma^N (2\pi)^{N/2}}, \tag{17}$$

$$\forall j \in [n + p], p(x_j^*|\alpha_j) = \frac{\exp\left(-(x_j^*)^2/(2\alpha_j)\right)}{\sqrt{2\pi\alpha_j}} \tag{18}$$

where the latter is a Gaussian prior on the elements of $\boldsymbol{x^*}$ with unknown variances $\{\alpha_j\}_{j=1}^{n+p}$ that are inferred on the fly from $\boldsymbol{y}, \boldsymbol{H}$. The details of the EM algorithm to jointly infer the signal vectors and their unknown element-wise variances, via maximizing the posterior probability $p(\boldsymbol{x^*}|\boldsymbol{y})$, can be found in Wipf & Rao (2004).

We refer to our algorithm A-HTP henceforth as $\ell_2$-HTP to distinguish it from $\ell_1$-HTP. We use CVXPY (Diamond & Boyd, 2016) to solve all the optimization problems, except for SBL which is implemented via EM. Also, note that by definition of the concept of permutation, we have a zero-sum constraint (referred to

as ZSC henceforth) on $e$ given by $\sum_{i=1}^{n} e_i = 0$. It is worth investigating whether this information helps to improve the estimate of $\boldsymbol{\beta}^*$. To this end, we incorporate this additional hard constraint in the Ar-Lasso and R-Lasso estimators, yielding two additional variants.

**Data generation:** In all the experiments, the entries of $\boldsymbol{A}$ and the non-zero values of $\boldsymbol{\beta}^*$ are sampled from $\mathcal{N}(0, 1)$. $\boldsymbol{P}_2$ is generated by randomly sampling from the family of $s$-sparse permutation matrices. The entries of $\boldsymbol{w}$ are independently sampled from $\mathcal{N}(0, \sigma^2)$ where $\sigma := f_r \times$ the mean absolute value of the entries of the noiseless measurement vector $\boldsymbol{PA\beta}^*$ with fraction $f_r \in (0, 1)$.

**Choice of parameters:** The regularization parameters $\lambda_\beta$ and $\lambda_e$ in Ar-Lasso, R-Lasso, Lasso and $\ell_1 - \ell_1$ algorithms are chosen through cross-validation on a held-out set of measurements. The unknown vectors $\boldsymbol{\beta}^*, \boldsymbol{e}^*$ are reconstructed using (say) 95% of the available measurements (in a set $\mathcal{R}$) using different values of $(\lambda_\beta, \lambda_e)$ from a set $\Lambda$. Let the signal reconstruction using regularization parameters $(l_1, l_2) \in \Lambda$ be called $\tilde{\boldsymbol{\beta}}_{l_1, l_2}, \tilde{\boldsymbol{e}}_{l_1, l_2}$. For each such estimate, the validation error $VE(l_1, l_2)$ is computed over the remaining 5% of the measurements forming a set $\mathcal{V}$. Here $VE(l_1, l_2) := \sum_{i \in \mathcal{V}} (y_i - \boldsymbol{A}^i \tilde{\boldsymbol{\beta}}_{l_1, l_2})^2$. Note that the set $\mathcal{V}$ contains only those measurements with known correspondences. The parameters $l_1, l_2$ which yield the least validation error are then selected. After this, the vectors $(\boldsymbol{\beta}^*, \boldsymbol{e}^*)$ are re-estimated using *all* measurements using the selected $(l_1, l_2)$ as regularization parameters. This technique is very effective because the validation error acts as a data-driven proxy for the immeasurable mean-squared error (MSE) as shown via confidence intervals in Zhang et al. (2014); Chetverikov et al. (2021). Note that our cross-validation approach does not require any prior 'training phase' on a class of similar signals to determine the optimal hyperparameters. Moreover Fig. 1(f) demonstrates that the RRMSE values remain stable over a decent range of values of $(\lambda_\beta, \lambda_e)$ as compared to those chosen by cross-validation.

Also note that $\ell_1$-Htp and $\ell_2$-Htp require the knowledge of $(k, s)$, which are typically unknown in practice but can be chosen via cross-validation. In our experiments, we observe that cross-validation overestimates $(k, s)$ by a factor of 2. Hence, we directly set $(k, s)$ to twice of their true value in the $\ell_1$-Htp and $\ell_2$-Htp algorithms. We select the learning rate in $\ell_1$-Htp and $\ell_2$-Htp through cross-validation. The number of iterations in $\ell_1$-Htp is set to 200, and that in $\ell_2$-Htp is set to 100. We always observed convergence within these iteration counts.

**Evaluation metric:** The evaluation metric used for all simulations is the relative root mean squared error (RRMSE) $\frac{\|\tilde{\boldsymbol{\beta}} - \boldsymbol{\beta}^*\|_2}{\|\boldsymbol{\beta}^*\|_2}$, where $\tilde{\boldsymbol{\beta}}$ is an estimate of $\boldsymbol{\beta}^*$. We report the RRMSE averaged over 50 randomly chosen instances of $\boldsymbol{P}_2$, with each permutation instance averaged over 50 random instances of measurement noise $\boldsymbol{w}$.

**Results:** In Fig. 1(a), we have plotted the RRMSE as a function of the number of measurements $N$ for $p = 240$, $k = 14$, $m = 32$, $s = 16$, and 2% noise (i.e. $f_r = 0.02$). We note that the approaches Ar-Lasso and $\ell_2$-Htp, which utilize the information of known correspondences, result in lower errors. $\ell_2$-Htp results in the least errors, followed by Ar-Lasso. $\ell_1 - \ell_1$ clearly produces much higher errors. Incorporation of ZSC in Ar-Lasso and R-Lasso only marginally improves the estimate of $\boldsymbol{\beta}^*$. As $N$ increases, $\ell_1$-Htp starts performing better than Ar-Lasso, indicating diminishing utility of known correspondences at a sufficiently large $N$.

In Fig. 2, histograms of the RRMSE values are displayed for $\ell_1$-Htp and $\ell_2$-Htp. Note that the estimation error depends on the specific (randomly chosen) instance of $\boldsymbol{P}_2$. We observe that $\ell_1$-Htp results in large errors many times. Moreover, we note that by using known correspondences in $\ell_2$-Htp, the variance of the RRMSE distribution significantly decreases for a given $N$. This is particularly prominent given fewer measurements, as seen in the case of $N \in \{80, 90\}$.

In Fig. 3, we display the scatter plot of the errors obtained on the randomly chosen permutation matrices averaged over 50 random noise instances as a function of $\|\boldsymbol{e}^*\|_1$ for Ar-Lasso and R-Lasso. Observe that for $N = 100$, the term $\|\boldsymbol{y}_1 - \boldsymbol{A}_1 \boldsymbol{\beta}\|_2^2$ in Ar-Lasso ensures that the errors do not depend much on the permutation error magnitude $\|\boldsymbol{e}^*\|_1$. On the other hand, without the information of known correspondences in R-Lasso, the scatter plot shows a clear positive correlation with $\|\boldsymbol{e}^*\|_1$. This correlation, however, dies down as we get more measurements, as seen in the case $N = 120$.

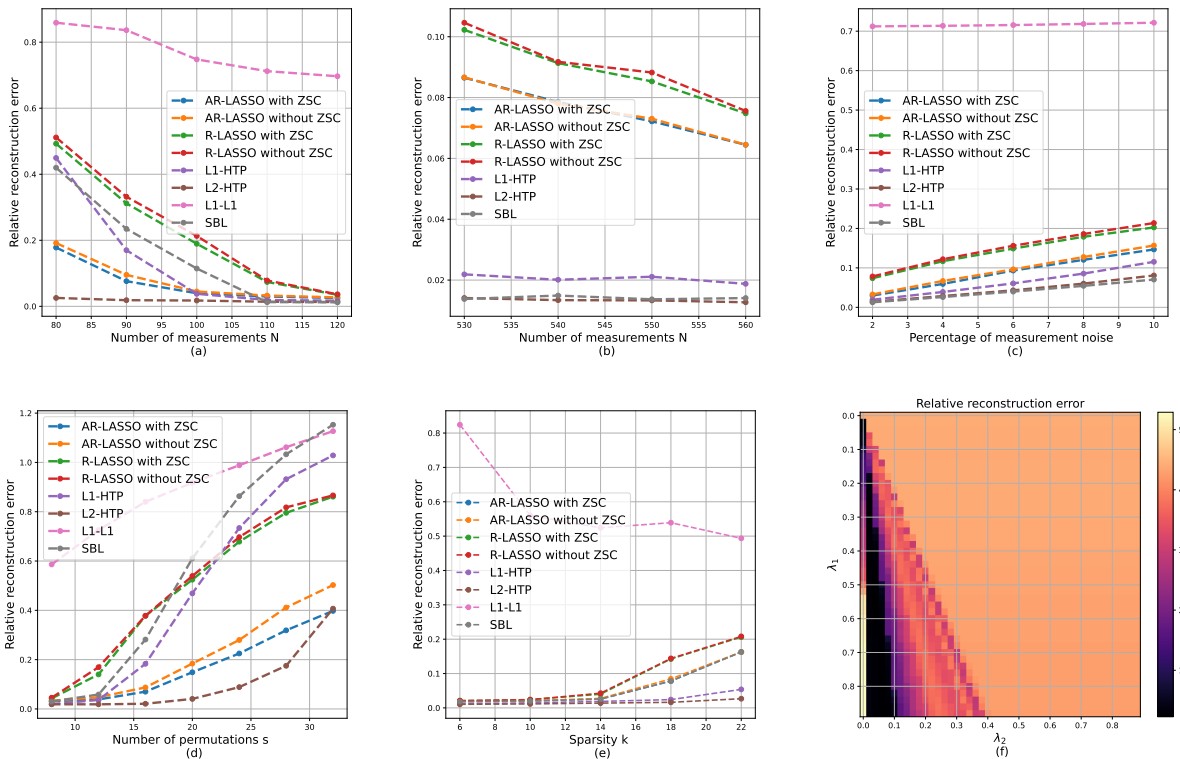

Figure 1: A comparison of the algorithms for unlabeled compressive sensing problem: The error plots as a function of (a) the number of measurements $N$ for $p = 240, k = 14, m = 32, s = 16, f_r = 0.02$; (b) the number of measurements $N$ in very high dimensions, i.e. $p = 1200, k = 70, m = 60, s = 80, f_r = 0.02$ (the errors with $\ell_1 - \ell_1$ are much higher than for other estimators, and are not shown here) (c) the noise level in terms of $f_r$ for $p = 240, N = 110, k = 14, m = 32, s = 16$; (d) the permutation level for $p = 240, N = 90, k = 14, m = 32, f_r = 0.02$; and (e) the sparsity level for $p = 240, N = 110, m = 32, s = 16, f_r = 0.02$. These plots show that using known correspondences as in $\ell_2$-HTP and AR-LASSO, results in a lower reconstruction error. *In plots (b), (c) and (e), the errors for some pairs of algorithms nearly overlap: AR-LASSO with and without the ZSC, R-LASSO with and without the ZSC, L2-HTP and SBL.* Plot (f) shows that the RRMSE is not significantly impacted by varying $\lambda_e, \lambda_\beta$ over a decent range of values as compared to the values chosen by cross-validation (see dark regions of the sub-figure).

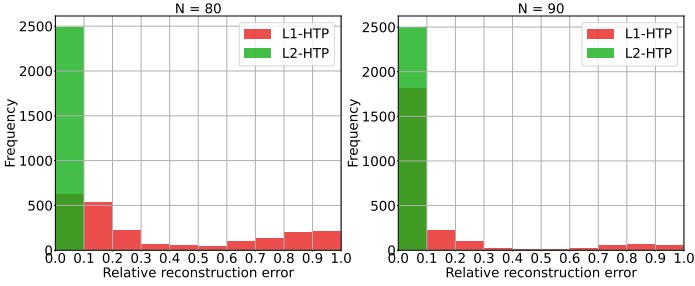

Figure 2: Distribution of the relative root mean squared error (RRMSE) values obtained using $\ell_1$-HTP and $\ell_2$-HTP for different instances of permutation matrix and measurement noise for $N = 80$ (left) and $N = 90$ (right) measurements. Utilizing the prior known correspondences (as done by $\ell_2$-HTP) results in a much lower standard deviation of the errors.

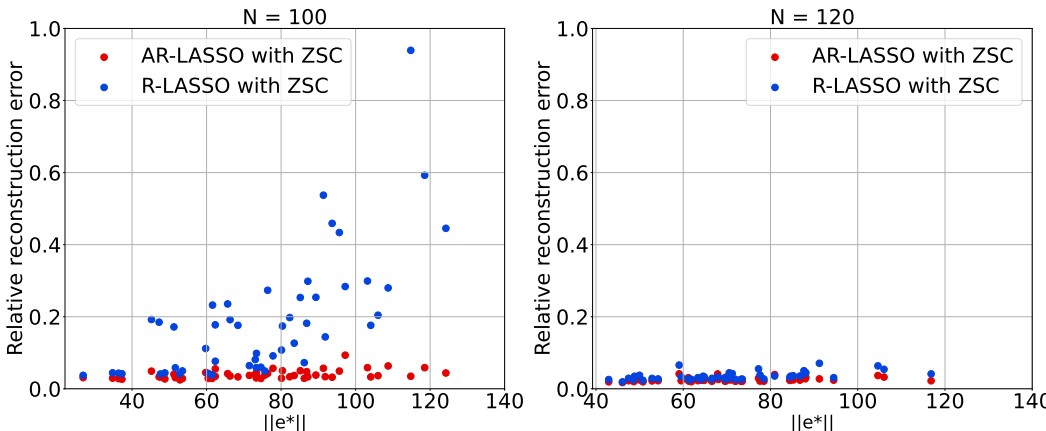

Figure 3: Scatter plots of RRMSE and permutation error magnitude $\|\boldsymbol{e}^*\|_1$ for $N = 100$ (left) and $N = 120$ (right) measurements. For $N = 100$, note the strong correlation of the estimation error with $\|\boldsymbol{e}^*\|_1$ when known correspondences are not utilized as in R-Lasso.

In Fig. 1(b), we compare the algorithms for different $N$ in higher dimensions, that is, for $p = 1200$, $k = 70$, $m = 60$, $s = 80$, and 2% measurement noise (i.e. $f_r = 0.02$).

Next, in Fig. 1(c), we compare the algorithms for different noise levels (i.e., different values of $f_r$) for $p = 240$, $N = 110$, $k = 14$, $m = 32$, and $s = 16$. We observe that for the Lasso-based approaches and $\ell_2$-Htp, the estimation error scales linearly with $\sigma$ as we expect from the upper bounds in Theorem 1 and Theorem 2.

In Fig. 1(d), we show errors for different permutation levels for $p = 240$, $N = 90$, $k = 14$, $m = 32$ and 2% measurement noise. Recall that the error upper bound for Ar-Lasso scales as $\sqrt{s}$. We note that the algorithms that utilize known correspondences are more robust in terms of the number of permutations in the measurements.

In Fig. 1(e), we show errors for different sparsity levels for $p = 240$, $N = 110$, $m = 32$, $s = 16$ and 2% measurement noise. During data generation for this experiment, we maintain that $48 \leq \|\boldsymbol{e}^*\|_1 \leq 52$ for all the 50 permutation instances, so that the results reflect the effect of changing sparsity level only. Note that the reconstruction error greatly depends on the permutation error $\|\boldsymbol{e}^*\|_1$.

**Performance of SBL**: The plots demonstrate that though SBL has shown excellent empirical performance in compressed sensing (Marques et al., 2018), it generally did not outperform Ar-Lasso or Ar-Htp for the unlabeled compressed sensing problem. A possible reason for this could be the assumption of zero-mean Gaussian priors on the elements of both $\boldsymbol{\beta}^*$ and $\boldsymbol{e}^*$, even though the elements of $\boldsymbol{e}^*$ can attain significantly large values.

**Performance of the $\ell_1 - \ell_1$ estimator:** In the results in Fig. 1, we notice that the $\ell_1 - \ell_1$ algorithm performs significantly worse than other methods. This is because, it uses an $\ell_1$ data fidelity term which assumes a Laplacian distribution on the noise in $\boldsymbol{y}$. However the noise in $\boldsymbol{y}$ contains a Gaussian component $\boldsymbol{w}$ and the component $\boldsymbol{e}^*$ due to permutations. The $\ell_1$ fidelity term ignores the Gaussian distribution of $\boldsymbol{w}$. Moreover, $\boldsymbol{e}^*$ does not necessarily follow the Laplacian distribution. The Ar-Lasso or R-Lasso estimators use the Gaussian distribution of the noise $\boldsymbol{w}$ and instead apply an $\ell_1$ sparsity-promoting penalty on the permutation error $\boldsymbol{e}^*$. It is important to note that the error bounds for Lasso and Ar-Lasso do not require that $\boldsymbol{e}^*$ follows a Laplacian distribution.

**Use of only permutation-free measurements:** Another natural baseline is a technique that uses *only* the $m$ permutation-free measurements for the sparse regression. This baseline is nothing but the regular Lasso and typically requires $m$ to be $O(k \log p)$. Having such a large number of permutation-free measurements

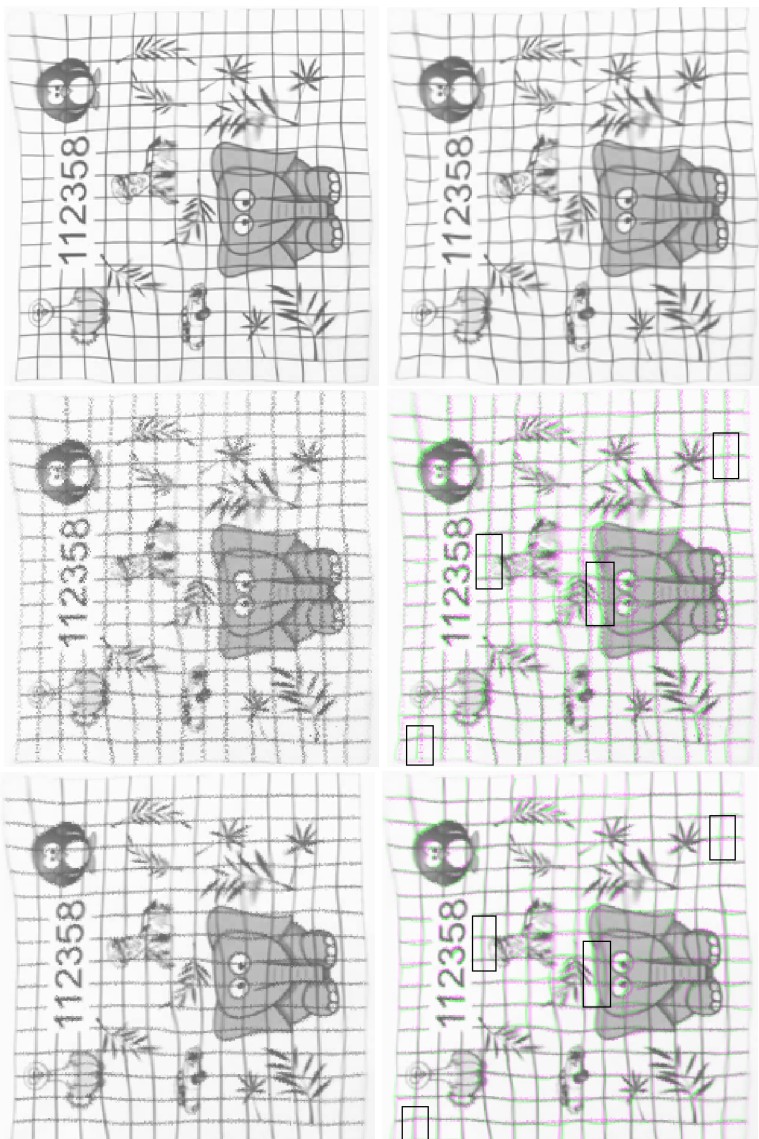

Figure 4: Row 1: Two refracted versions $I$ (left) and $R$ (right) generated synthetically by warping reference image $Z$ using a displacement vector field created using sparse random linear combination of 2D-DCT basis vectors. Row 2: The reconstruction of $R$ using R-Lasso, i.e. $R_{\text{R-Lasso}}$ (left) and the overlap between $R$ and $R_{\text{R-Lasso}}$ (right). Row 3: The reconstruction of $R$ using Ar-Lasso, i.e. $R_{\text{Ar-Lasso}}$ (left) and the overlap between $R$ and $R_{\text{Ar-Lasso}}$ (right). Notice the significantly better overlap for Ar-Lasso than for R-Lasso. For easier visualization, we highlight the regions of inaccurate overlap between $R$ and $R_{\text{R-Lasso}}$ (row 2, right) and the more accurate overlap between $R$ and $R_{\text{Ar-Lasso}}$ (row 3, right), using black boxes. The convention followed for display of overlaps is described under 'Visualization of Results' in Sec. 5.1.

may however be unrealistic. Our experimental results show that the RRMSE values with such a baseline can be more than 0.7 for small $m$, and hence are not included here.

Average compute time values of our algorithms for unlabeled compressive sensing with known correspondences are presented in the supplemental material.

## 5.1 Application of Unlabeled Compressive Sensing in Image Registration

Consider a scene submerged under a wavy water surface when imaged using a camera in air. The images of such a scene contains distinct non-rigid deformations when compared to an image $Z$ of the same scene if the scene were located in air. Consider two such refracted images denoted by $I$ and $R$, corresponding to different positions of the wavy water surface. It is known that the deformation vector field between images $I$ and $R$ can be expressed as sparse linear combination of the 2D discrete cosine transform (DCT) basis vectors (see James et al. (2019); Wulff & Black (2015)). Let both $I$ and $R$ be of size $h_1 \times h_2$. Define $\boldsymbol{u_1} \in \mathbb{R}^{h_1 h_2}$ and $\boldsymbol{u_2} \in \mathbb{R}^{h_1 h_2}$ to be the vectorized displacement fields from $I$ to $R$ in the horizontal and vertical directions respectively. Let $\boldsymbol{B} \in \mathbb{R}^{h_1 h_2 \times h_1 h_2}$ denote the 2D DCT matrix. Then, we can represent the displacement fields as follows:

$$\boldsymbol{u_1} = \boldsymbol{B}\boldsymbol{\theta_1}, \boldsymbol{u_2} = \boldsymbol{B}\boldsymbol{\theta_2}, \tag{19}$$

where $\boldsymbol{\theta_1}, \boldsymbol{\theta_2} \in \mathbb{R}^{h_1 h_2}$ are unknown sparse coefficient vectors. In a similar work, Sresth et al. (2024) assumes that the displacement vector field is band-limited, that is, they assume that $\boldsymbol{B}$ consists of only some $d \ll h_1 h_2$ *low-frequency* 2D DCT basis column vectors. This assumption may not always hold and hence in our work, the matrix $\boldsymbol{B}$ consists of all $h_1 h_2$ 2D DCT basis vectors, and accordingly, $(\boldsymbol{\theta_1}, \boldsymbol{\theta_2})$ are sparse vectors whose support is not restricted to just the *low-frequency* 2D DCT coefficients.

Obtaining the displacement vector fields from observed images is a difficult problem. Conventional optical flow algorithms do not leverage the property of bandlimitedness or sparsity of the unknown $\boldsymbol{u_1}, \boldsymbol{u_2}$ in the 2D DCT basis. However, we can obtain the displacement vector information at a *subset* of pixels $\mathcal{H}$ in $I$ using salient feature point matching techniques such as SIFT (scale invariant feature transform) (Lowe, 2004), Harris corner detectors Harris & Stephens (1988) or SURF (speeded up robust features) Bay et al. (2006). Given corresponding point pairs $\{(\boldsymbol{p_{I,k}}, \boldsymbol{p_{R,k}})\}_{k=1}^{|\mathcal{H}|}$ in images $I$ and $R$, the displacement at the $k^{\text{th}}$ point can be computed as $(u_{1k}, u_{2k}) := \boldsymbol{p_{R,k}} - \boldsymbol{p_{I,k}}$. Note that each $\boldsymbol{p_{I,k}}$ and $\boldsymbol{p_{R,k}}$ is a 2D vector. The topic of salient feature point matching is well researched in the computer vision literature, and so such point pairs provide useful information. Given this set $\mathcal{H}$, we have $\boldsymbol{u_{1,\mathcal{H}}} = \boldsymbol{B_{\mathcal{H}}}\boldsymbol{\theta_1}, \boldsymbol{u_{2,\mathcal{H}}} = \boldsymbol{B_{\mathcal{H}}}\boldsymbol{\theta_2}$. Here, $\boldsymbol{B_{\mathcal{H}}} \in \mathbb{R}^{|\mathcal{H}| \times d}$ contains the rows from $\boldsymbol{B}$ corresponding to only those pixel locations which belong to $\mathcal{H}$, and $\boldsymbol{u_{1,\mathcal{H}}}, \boldsymbol{u_{2,\mathcal{H}}} \in \mathbb{R}^{|\mathcal{H}|}$ are sub-vectors of $\boldsymbol{u_1}, \boldsymbol{u_2}$ respectively containing displacement values only from locations in $\mathcal{H}$. The task now is to estimate $\boldsymbol{\theta_1}, \boldsymbol{\theta_2}$ given $\mathcal{H}$ and $\boldsymbol{u_{1,\mathcal{H}}}, \boldsymbol{u_{2,\mathcal{H}}}$. It is clear that this is a compressive sensing problem. However it is also an *unlabeled* compressive sensing problem, because it is well known that salient feature point matching algorithms can produce a small number of erroneous point matches, especially if the images contain self-similarity. That is, if the images contain structurally similar patches in different locations, then some points from one region can be erroneously matched to those in other regions because the point matching techniques leverage mostly local information. Hence, we can write the following:

$$\boldsymbol{u_{1,\mathcal{H}}} = \boldsymbol{B_{\mathcal{H}}}\boldsymbol{\theta_1} + \boldsymbol{\delta_1} + \boldsymbol{\eta_{1,\mathcal{H}}}; \boldsymbol{u_{2,\mathcal{H}}} = \boldsymbol{B_{\mathcal{H}}}\boldsymbol{\theta_2} + \boldsymbol{\delta_2} + \boldsymbol{\eta_{2,\mathcal{H}}}, \tag{20}$$

where $\boldsymbol{\eta_{1,\mathcal{H}}} \in \mathbb{R}^{|\mathcal{H}|}, \boldsymbol{\eta_{2,\mathcal{H}}} \in \mathbb{R}^{|\mathcal{H}|}$ represent small errors in the point coordinates modeled as zero-mean Gaussian random variables with small variance, and $\boldsymbol{\delta_1} \in \mathbb{R}^{|\mathcal{H}|}, \boldsymbol{\delta_2} \in \mathbb{R}^{|\mathcal{H}|}$ represent sparse vectors whose non-zero elements represents the permutation errors. Furthermore, in motion estimation problems such as the one under consideration, we can also use a small number of expert (accurate) point-pair annotations as side information. Let us denote this set of accurate point-pair matches by $\mathcal{S}$. Using notation similar to that in equation 20, we have:

$$\boldsymbol{u_{1,\mathcal{S}}} = \boldsymbol{B_{\mathcal{S}}}\boldsymbol{\theta_1} + \boldsymbol{\eta_{1,\mathcal{S}}}; \boldsymbol{u_{2,\mathcal{S}}} = \boldsymbol{B_{\mathcal{S}}}\boldsymbol{\theta_2} + \boldsymbol{\eta_{2,\mathcal{S}}}. \tag{21}$$

Note that there are no terms for permutation errors in equation 21 as these are expert annotations.

**Motion Estimator:** Considering both equation 21 and equation 20, we have an AR-LASSO based estimator which minimizes the following cost functions with respect to $(\boldsymbol{\theta_1}, \boldsymbol{\delta_1})$ and $(\boldsymbol{\theta_2}, \boldsymbol{\delta_2})$ respectively:

$$J_1(\boldsymbol{\theta_1}, \boldsymbol{\delta_1}) := \|\boldsymbol{u_{1,\mathcal{H}}} - \boldsymbol{B_{\mathcal{H}}}\boldsymbol{\theta_1} - \boldsymbol{\delta_1}\|_2^2 + \|\boldsymbol{u_{1,\mathcal{S}}} - \boldsymbol{B_{\mathcal{S}}}\boldsymbol{\theta_1}\|_2^2 + \lambda_1\|\boldsymbol{\theta_1}\|_1 + \lambda_2\|\boldsymbol{\delta_1}\|_1, \tag{22}$$

$$J_2(\boldsymbol{\theta_2}, \boldsymbol{\delta_2}) := \|\boldsymbol{u_{2,\mathcal{H}}} - \boldsymbol{B_{\mathcal{H}}}\boldsymbol{\theta_2} - \boldsymbol{\delta_2}\|_2^2 + \|\boldsymbol{u_{2,\mathcal{S}}} - \boldsymbol{B_{\mathcal{S}}}\boldsymbol{\theta_2}\|_2^2 + \lambda_1\|\boldsymbol{\theta_2}\|_1 + \lambda_2\|\boldsymbol{\delta_2}\|_1, \tag{23}$$

where $\lambda_1 > 0, \lambda_2 > 0$ are regularization parameters. Given $\boldsymbol{\theta_1}$ and $\boldsymbol{\theta_2}$, the displacement vector fields can be obtained using equation 19, and then applied to $I$ in order to align it with $R$.

**Experiment:** Consider two grayscale images $I$ and $R$ from row 1 of Fig. 4 respectively. The image $I$ was chosen arbitrarily as the reference image. Image $R$ was generated synthetically by warping $I$ using a displacement vector field that was sparse in the 2D-DCT basis (for both $X$ and $Y$ components of the motion) following the model in equation 19. To obtain $\boldsymbol{\theta_1}, \boldsymbol{\theta_2}$ and thus determine the motion from $I$ to $R$, we obtain a set $\mathcal{H}$ of $n = 452$ salient feature point-pairs in the images $I$ and $R$ by combining the outputs of SIFT (Lowe, 2004), SURF (Bay et al., 2006) and Harris descriptors (Harris & Stephens, 1988). Furthermore, we accurately annotate a set $\mathcal{S}$ of $m = 18$ corresponding point-pairs in the images $I$ and $R$. The correspondences of the $m$ point-pairs in $\mathcal{S}$ are known *accurately*, whereas a small number of correspondences in the point-pairs in $\mathcal{H}$ may be incorrect due to errors in point matching algorithms. For example, the different grid corners from the two images (where vertical and horizontal lines of the grid intersect) in row 1 of Fig. 4 can be incorrectly matched with each other, as point matching algorithms rely on local similarity features. Note that the indices of the erroneous point-pairs in $\mathcal{H}$ are unknown. We can use the AR-LASSO algorithm to estimate $\boldsymbol{\theta_1}, \boldsymbol{\theta_2}$ given $\mathcal{H}, \mathcal{S}$ and thus reconstruct $R$ by warping the image $I$ using displacement fields $\boldsymbol{u_1} := \boldsymbol{B\theta_1}, \boldsymbol{u_2} := \boldsymbol{B\theta_2}$. To show the utility of $m$ known correspondences, we also perform the reconstruction of $R$ using $\boldsymbol{\theta_1}, \boldsymbol{\theta_2}$ obtained from R-LASSO. We refer to the reconstructions of $R$ using AR-LASSO and R-LASSO as $R_{\text{AR-LASSO}}$ and $R_{\text{R-LASSO}}$ respectively.

**Visualization of Results:** We display the reference image $I$, its underwater refraction $R$, the reconstructions of $R$, i.e. $R_{\text{AR-LASSO}}$ and $R_{\text{R-LASSO}}$, and the overlap between $R$ and their reconstructions (i.e., $R_{\text{AR-LASSO}}$ and $R_{\text{R-LASSO}}$) in Fig. 4. To display the overlap between two images $R$ and its estimate $R_{\text{AR-LASSO}}$, we create a color (RGB) image, and set the Red channel to $R$, the Green channel to $R_{\text{AR-LASSO}}$ and the Blue channel to all zeros. If the images $R$ and $R_{\text{AR-LASSO}}$ are well aligned, the resulting RGB image will appear mostly gray. In case, there is misalignment, the RGB image will show green or magenta shades in different places. Similarly, the overlap between $R$ and $R_{\text{R-LASSO}}$ is also displayed. Clearly, the overlap between $R$ and $R_{\text{AR-LASSO}}$ is more accurate than that between $R$ and $R_{\text{R-LASSO}}$ as evidenced by the larger proportion of green or magenta regions in the latter (eg: elephant's head). Note that $R_{\text{R-LASSO}}$ also looks noisy. The relative reconstruction error of the reconstructed image $R$ using AR-LASSO is 0.141 while that using R-LASSO is 0.167. The relative reconstruction error of the displacement field $(\boldsymbol{u}_1, \boldsymbol{u}_2)$ using AR-LASSO is 0.243 while that using R-LASSO is 0.343.

These results show a proof of concept of the utility of AR-LASSO in image alignment.

## 6 Conclusion

We studied the problem of unlabeled compressive sensing with the assumption that the regression vector is sparse and given additional knowledge of a few correspondences. We proposed a tractable LASSO-based estimator and derived theoretical performance bounds for our algorithm. We also presented another estimator based on a modified form of Hard Thresholding Pursuit, with theoretical analysis. We verified the theoretical findings through numerical experiments. We compared our algorithm with a hard thresholding approach and an $\ell_1$ norm formulation and demonstrated that our algorithm outperforms them. Additionally, we illustrated that having information about a small number of accurate correspondences reduces the sensitivity of estimation error on the severity of permutation corruption. Lastly, we demonstrated a practical application of our framework in image registration.

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

# A   Appendix

Please refer to supplemental material for additional experiments and the proofs of lemmas and theorems.

