# Supplementary Material for TMLR Submission: unlabeled Compressive Sensing under Sparse Permutation and Prior Information

**Garweet Sresth**  *garweetsresth@gmail.com*
*Department of Electrical Engineering*
*IIT Bombay*

**Satish Mulleti**  *satish.mulleti@iitb.ac.in*
*Department of Electrical Engineering*
*IIT Bombay*

**Ajit Rajwade**  *ajitvr@cse.iitb.ac.in*
*Department of Computer Science and Engineering*
*IIT Bombay*

**Reviewed on OpenReview:** *https://openreview.net/forum?id=HaAg9RN7Hi*

## 1 Proof of Theorem 1

*Proof.* Referring to the notation in the main paper, we have

$$\boldsymbol{y}_1 = \boldsymbol{A}_1\boldsymbol{\beta^*} + \boldsymbol{w_1}, \tag{1}$$

$$\boldsymbol{y}_2 = \boldsymbol{A}_2\boldsymbol{\beta^*} + \sqrt{n}\boldsymbol{e^*} + \boldsymbol{w_2}. \tag{2}$$

Next, because $(\tilde{\boldsymbol{\beta}}, \tilde{\boldsymbol{e}})$ minimize $L(\boldsymbol{\beta}, \boldsymbol{e})$, we have $L(\tilde{\boldsymbol{\beta}}, \tilde{\boldsymbol{e}}) \leq L(\boldsymbol{\beta^*}, \boldsymbol{e^*})$, that is

$$\frac{1}{2N}\|\boldsymbol{y_1} - \boldsymbol{A_1}\tilde{\boldsymbol{\beta}}\|_2^2 + \frac{1}{2N}\|\boldsymbol{y_2} - \boldsymbol{A_2}\tilde{\boldsymbol{\beta}} - \sqrt{n}\tilde{\boldsymbol{e}}\|_2^2 + \lambda_\beta\|\tilde{\boldsymbol{\beta}}\|_1 + \lambda_e\|\tilde{\boldsymbol{e}}\|_1$$
$$\leq \frac{1}{2N}\|\boldsymbol{y_1} - \boldsymbol{A_1}\boldsymbol{\beta^*}\|_2^2 + \frac{1}{2N}\|\boldsymbol{y_2} - \boldsymbol{A_2}\boldsymbol{\beta^*} - \sqrt{n}\boldsymbol{e^*}\|_2^2 + \lambda_\beta\|\boldsymbol{\beta^*}\|_1 + \lambda_e\|\boldsymbol{e^*}\|_1. \tag{3}$$

Let $\boldsymbol{h} := \tilde{\boldsymbol{\beta}} - \boldsymbol{\beta^*}$ and $\boldsymbol{f} := \tilde{\boldsymbol{e}} - \boldsymbol{e^*}$. Then, we have

$$\frac{1}{2N}\|\boldsymbol{y_1} - \boldsymbol{A_1}\boldsymbol{\beta^*} - \boldsymbol{A_1}\boldsymbol{h}\|_2^2 + \frac{1}{2N}\|\boldsymbol{y_2} - \boldsymbol{A_2}\boldsymbol{\beta^*} - \sqrt{n}\boldsymbol{e^*} - \boldsymbol{A_2}\boldsymbol{h} - \sqrt{n}\boldsymbol{f}\|_2^2 + \lambda_\beta\|\tilde{\boldsymbol{\beta}}\|_1 + \lambda_e\|\tilde{\boldsymbol{e}}\|_1$$
$$\leq \frac{1}{2N}\|\boldsymbol{y_1} - \boldsymbol{A_1}\boldsymbol{\beta^*}\|_2^2 + \frac{1}{2N}\|\boldsymbol{y_2} - \boldsymbol{A_2}\boldsymbol{\beta^*} - \sqrt{n}\boldsymbol{e^*}\|_2^2 + \lambda_\beta\|\boldsymbol{\beta^*}\|_1 + \lambda_e\|\boldsymbol{e^*}\|_1. \tag{4}$$

Next, we use equation 1 and equation 2 to obtain

$$\frac{1}{2N}\|\boldsymbol{w_1} - \boldsymbol{A_1}\boldsymbol{h}\|_2^2 + \frac{1}{2N}\|\boldsymbol{w_2} - (\boldsymbol{A_2}\boldsymbol{h} + \sqrt{n}\boldsymbol{f})\|_2^2 + \lambda_\beta\|\tilde{\boldsymbol{\beta}}\|_1 + \lambda_e\|\tilde{\boldsymbol{e}}\|_1$$
$$\leq \frac{1}{2N}\|\boldsymbol{w_1}\|_2^2 + \frac{1}{2N}\|\boldsymbol{w_2}\|_2^2 + \lambda_\beta\|\boldsymbol{\beta^*}\|_1 + \lambda_e\|\boldsymbol{e^*}\|_1 \tag{5}$$

or

$$\frac{1}{2N}\|\boldsymbol{A_1}\boldsymbol{h}\|_2^2 - \frac{1}{N}\boldsymbol{w_1^T}\boldsymbol{A_1}\boldsymbol{h} + \frac{1}{2N}\|\boldsymbol{A_2}\boldsymbol{h} + \sqrt{n}\boldsymbol{f}\|_2^2 - \frac{1}{N}\boldsymbol{w_2^T}(\boldsymbol{A_2}\boldsymbol{h} + \sqrt{n}\boldsymbol{f})$$
$$\leq \lambda_\beta(\|\boldsymbol{\beta^*}\|_1 - \|\tilde{\boldsymbol{\beta}}\|_1) + \lambda_e(\|\boldsymbol{e^*}\|_1 - \|\tilde{\boldsymbol{e}}\|_1). \tag{6}$$

Following standard notation, the vector $\boldsymbol{h_T}$ stands for a copy of vector $\boldsymbol{h}$ such that $\forall i \in T, h_T(i) = h(i)$ and $\forall i \notin T, h_T(i) = 0$. Next, we upper bound the term $(\|\boldsymbol{\beta^*}\|_1 - \|\tilde{\boldsymbol{\beta}}\|_1)$ as follows:

$$\|\boldsymbol{\beta^*}\|_1 - \|\tilde{\boldsymbol{\beta}}\|_1 = \|\boldsymbol{\beta^*}\|_1 - \|\boldsymbol{h} + \boldsymbol{\beta^*}\|_1 = \|\boldsymbol{\beta^*}\|_1 - \|\boldsymbol{h_T} + \boldsymbol{\beta^*}\|_1 - \|\boldsymbol{h_{T^C}}\|_1 \le \|\boldsymbol{h_T}\|_1 - \|\boldsymbol{h_{T^C}}\|_1. \tag{7}$$

The last step uses the reverse triangle inequality. Similarly, we also have

$$\|\boldsymbol{e^*}\|_1 - \|\tilde{\boldsymbol{e}}\|_1 \le |\boldsymbol{f_S}\|_1 - \|\boldsymbol{f_{S^C}}\|_1. \tag{8}$$

From equation 6, we have the following using Hölder's inequality as well as equation 7 and equation 8:

$$\frac{1}{2N}\|\boldsymbol{A_1 h}\|_2^2 + \frac{1}{2N}\|\boldsymbol{A_2 h} + \sqrt{n}\boldsymbol{f}\|_2^2 \le \frac{1}{N}\boldsymbol{w_1}^T \boldsymbol{A_1 h} + \frac{1}{N}\boldsymbol{w_2}^T(\boldsymbol{A_2 h} + \sqrt{n}\boldsymbol{f}) + \lambda_\beta(\|\boldsymbol{\beta^*}\|_1 - \|\tilde{\boldsymbol{\beta}}\|_1) + \lambda_e(\|\boldsymbol{e^*}\|_1 - \|\tilde{\boldsymbol{e}}\|_1),$$

$$= \frac{1}{N}(\boldsymbol{A}^T \boldsymbol{w})^T \boldsymbol{h} + \frac{\sqrt{n}}{N}\boldsymbol{w_2}^T \boldsymbol{f} + \lambda_\beta(\|\boldsymbol{\beta^*}\|_1 - \|\tilde{\boldsymbol{\beta}}\|_1) + \lambda_e(\|\boldsymbol{e^*}\|_1 - \|\tilde{\boldsymbol{e}}\|_1)$$

$$\le \frac{1}{N}\|\boldsymbol{A}^T \boldsymbol{w}\|_\infty\|\boldsymbol{h}\|_1 + \frac{\sqrt{n}}{N}\|\boldsymbol{w_2}\|_\infty\|\boldsymbol{f}\|_1 + \lambda_\beta(\|\boldsymbol{h_T}\|_1 - \|\boldsymbol{h_{T^C}}\|_1) + \lambda_e(\|\boldsymbol{f_S}\|_1 - \|\boldsymbol{f_{S^C}}\|_1). \tag{9}$$

Next, we have that

$$\frac{1}{2N}\|\boldsymbol{A_1 h}\|_2^2 + \frac{1}{2N}\|\boldsymbol{A_2 h} + \sqrt{n}\boldsymbol{f}\|_2^2 \le \|\boldsymbol{h_T}\|_1\left(\lambda_\beta + \frac{1}{N}\|\boldsymbol{A}^T \boldsymbol{w}\|_\infty\right) + \|\boldsymbol{h_{T^C}}\|_1\left(-\lambda_\beta + \frac{1}{N}\|\boldsymbol{A}^T \boldsymbol{w}\|_\infty\right)$$

$$+ \|\boldsymbol{f_S}\|_1\left(\lambda_e + \frac{\sqrt{n}}{N}\|\boldsymbol{w_2}\|_\infty\right) + \|\boldsymbol{f_{S^C}}\|_1\left(-\lambda_e + \frac{\sqrt{n}}{N}\|\boldsymbol{w_2}\|_\infty\right). \tag{10}$$

Next, choose $\lambda_\beta, \lambda_e$ such that $\frac{1}{N}\|\boldsymbol{A}^T \boldsymbol{w}\|_\infty \le \frac{\lambda_\beta}{2}$ and $\frac{\sqrt{n}}{N}\|\boldsymbol{w_2}\|_\infty \le \frac{\lambda_e}{2}$. Say $\lambda_\beta = \frac{2}{\rho}\frac{\|\boldsymbol{A}^T \boldsymbol{w}\|_\infty}{N}$ and $\lambda_e = \frac{2\sqrt{n}}{N}\|\boldsymbol{w_2}\|_\infty$. $\rho \in (0,1)$ is a positive constant which controls the sparsity level in the regression vector and sparse error vector. If a large number of permutations is expected, then a smaller value of $\rho$ should be used and vice-versa. Now, from equation 10, and using the bounds involving $\lambda_\beta, \lambda_e$, we have that

$$\frac{1}{2N}\|\boldsymbol{A_1 h}\|_2^2 + \frac{1}{2N}\|\boldsymbol{A_2 h} + \sqrt{n}\boldsymbol{f}\|_2^2 \le \frac{3\lambda_\beta}{2}\|\boldsymbol{h_T}\|_1 - \frac{\lambda_\beta}{2}\|\boldsymbol{h_{T^C}}\|_1 + \frac{3\lambda_e}{2}\|\boldsymbol{f_S}\|_1 - \frac{\lambda_e}{2}\|\boldsymbol{f_{S^C}}\|_1,$$

$$\le \frac{3\lambda_\beta}{2}\|\boldsymbol{h_T}\|_1 + \frac{3\lambda_e}{2}\|\boldsymbol{f_S}\|_1. \tag{11}$$

Note that the term on the left hand side in equation 11 is lower bounded by 0, and hence by definition of set $\mathcal{C}$ from Sec. 3.1 of the main paper, we have $(\boldsymbol{h}, \boldsymbol{f}) \in C$ with $\lambda = \frac{\lambda_e}{\lambda_\beta}$. Using Gaussian tail bounds, it can be shown that $\|\boldsymbol{A}^T \boldsymbol{w}\|_\infty \le 2\sigma\sqrt{\xi(\boldsymbol{\Sigma})N \log p}$ with probability at least $1 - 2/p$ and $\|\boldsymbol{w_2}\|_\infty \le 2\sigma\sqrt{\log n}$ with probability at least $1 - 2/n$. Plugging the values in the expressions for $\lambda_\beta$ and $\lambda_e$, we obtain $\lambda = \rho\sqrt{\frac{n}{N}\frac{\log n}{\xi(\boldsymbol{\Sigma})\log p}}$.

Now, since we have established that $(\boldsymbol{h}, \boldsymbol{f}) \in C$, we are ready to apply Lemma 3 (equation 24) with all the assumptions as mentioned in Lemma 3 to lower bound the term $\frac{1}{2N}\|\boldsymbol{A_1 h}\|_2^2 + \frac{1}{2N}\|\boldsymbol{A_2 h} + \sqrt{n}\boldsymbol{f}\|_2^2$ in equation 11. Doing so, we obtain

$$\left(\min\left(\frac{C_{\min}(\boldsymbol{\Sigma})}{512}, \frac{n}{32N}\right) - k_m \frac{n}{N}\right)(\|\boldsymbol{h}\|_2 + \|\boldsymbol{f}\|_2)^2 \le \frac{1}{2N}\|\boldsymbol{A_1 h}\|_2^2 + \frac{1}{2N}\|\boldsymbol{A_2 h} + \sqrt{n}\boldsymbol{f}\|_2^2 \le \frac{3\lambda_\beta}{2}\|\boldsymbol{h_T}\|_1 + \frac{3\lambda_e}{2}\|\boldsymbol{f_S}\|_1. \tag{12}$$

Since $\boldsymbol{h_T}$ is $k$-sparse and $\boldsymbol{f_S}$ is $s$-sparse, we use the norm-inequalities to get

$$\left(\min\left(\frac{C_{\min}(\boldsymbol{\Sigma})}{512}, \frac{n}{32N}\right) - k_m \frac{n}{N}\right)(\|\boldsymbol{h}\|_2 + \|\boldsymbol{f}\|_2)^2 \le \frac{3}{2}\lambda_\beta\|\boldsymbol{h_T}\|_1 + \frac{3}{2}\lambda_e\|\boldsymbol{f_S}\|_1 \le \frac{3}{2}\sqrt{k}\lambda_\beta\|\boldsymbol{h}\|_2 + \frac{3}{2}\sqrt{s}\lambda_e\|\boldsymbol{f}\|_2, \tag{13}$$

or

$$k_l^2(\|\boldsymbol{h}\|_2 + \|\boldsymbol{f}\|_2)^2 \leq \frac{3}{2} \max(\sqrt{k}\lambda_\beta, \sqrt{s}\lambda_e)(\|\boldsymbol{h}\|_2 + \|\boldsymbol{f}\|_2) \text{ where } k_l^2 := \left(\left(\frac{C_{\min}(\boldsymbol{\Sigma})}{512}, \frac{n}{32N}\right) - k_m \frac{n}{N}\right), \quad (14)$$

or

$$\begin{aligned}
(\|\boldsymbol{h}\|_2 + \|\boldsymbol{f}\|_2) &\leq \frac{3}{2} k_l^{-2} \max(\sqrt{k}\lambda_\beta, \sqrt{s}\lambda_e) \\
&= \frac{3}{2} k_l^{-2} \max\left(\frac{2\sqrt{k}}{\rho N}\|\boldsymbol{A}^T\boldsymbol{w}\|_\infty, \frac{2\sqrt{ns}}{N}\|\boldsymbol{w_2}\|_\infty\right).
\end{aligned} \quad (15)$$

Note that we have substituted the expressions for $\lambda_\beta, \lambda_e$ in the last step.

Using Gaussian tail bounds since the elements of $\boldsymbol{w_1}$ and $\boldsymbol{w_2}$ are Gaussian distributed, we now substitute expressions for $\lambda_\beta$ and $\lambda_e$ in the last step. Using the results that $\|\boldsymbol{A}^T\boldsymbol{w}\|_\infty \leq 2\sigma\sqrt{\xi(\boldsymbol{\Sigma})N \log p}$ with probability at least $1 - 2/p$ and $\|\boldsymbol{w_2}\|_\infty \leq 2\sigma\sqrt{\log n}$ with probability at least $1 - 2/n$, we obtain:

$$(\|\boldsymbol{h}\|_2 + \|\boldsymbol{f}\|_2) \leq 6\sigma k_l^{-2} \max\left(\frac{1}{\rho}\sqrt{\frac{\xi(\boldsymbol{\Sigma})k \log p}{N}}, \sqrt{\frac{n}{N}\frac{s \log n}{N}}\right). \quad (16)$$

By the complement of Boole's inequality, this occurs with probability greater than or equal to $1 - 2/p - 2/n$. $\qquad\square$

## 2 Lemmas

**Lemma 1** (**Generalised, extended, restricted eigen-value condition**). *Consider the Gaussian sensing matrix $\boldsymbol{A} \in \mathbb{R}^{N \times p}$ whose rows are i.i.d. $\mathcal{N}(\boldsymbol{0}, \boldsymbol{\Sigma})$ where $\boldsymbol{0}$ has $p$ elements and $\boldsymbol{\Sigma}$ is a $p \times p$ covariance matrix. We have the set $\mathcal{C} := \{(\boldsymbol{h}, \boldsymbol{f}) \in (\mathbb{R}^p \times \mathbb{R}^n) \text{ such that } \|\boldsymbol{h_{T^C}}\|_1 + \lambda\|\boldsymbol{f_{S^C}}\|_1 \leq 3\|\boldsymbol{h_T}\|_1 + 3\lambda\|\boldsymbol{f_S}\|_1\}$ as defined earlier. Select $\lambda = \rho\sqrt{\frac{n}{N}\frac{\log n}{\xi(\boldsymbol{\Sigma})\log p}}$, where $\rho \in (0,1)$ is a constant. If $s \leq c_1 \frac{N}{\rho^2 \log n}$ and $N \geq c_2 \frac{\xi(\boldsymbol{\Sigma})}{C_{min}(\boldsymbol{\Sigma})} k \log p$, then the following inequality holds with probability atleast $1 - c_3 \exp(-c_4 N)$:*

$$min\left(\frac{C_{min}(\boldsymbol{\Sigma})}{128}, \frac{n}{8N}\right)(\|\boldsymbol{h}\|_2^2 + \|\boldsymbol{f}\|_2^2) \leq \frac{1}{N}\|\boldsymbol{Ah}\|_2^2 + \frac{n}{N}\|\boldsymbol{f}\|_2^2 \ \forall \ (\boldsymbol{h}, \boldsymbol{f}) \in \mathcal{C}, \quad (17)$$

*where $c_1, c_2, c_3, c_4$ are positive constants.*

*Proof.* We lower bound the term $\frac{1}{\sqrt{N}}\|\boldsymbol{Ah}\|_2$ by using a concentration result from Raskutti et al. (2010). We get

$$\frac{\sqrt{C_{\min}(\boldsymbol{\Sigma})}}{4}\|\boldsymbol{h}\|_2 - 9\sqrt{\frac{\xi(\boldsymbol{\Sigma})\log p}{N}}\|\boldsymbol{h}\|_1 \leq \frac{1}{\sqrt{N}}\|\boldsymbol{Ah}\|_2 \text{ with probability greater than } 1 - c_1 \exp(-c_2 N), \quad (18)$$

where $c_1 > 0, c_2 > 0$ are some constants. Next, note that $\|\boldsymbol{h_{T^C}}\|_1 + \lambda\|\boldsymbol{f_{S^C}}\|_1 \leq 3\|\boldsymbol{h_T}\|_1 + 3\lambda\|\boldsymbol{f_S}\|_1\}$ implies $\|\boldsymbol{h}\|_1 \leq 4\|\boldsymbol{h_T}\|_1 + 3\lambda\|\boldsymbol{f_S}\|_1\} \leq 4\sqrt{k}\|\boldsymbol{h}\|_2 + 3\lambda\sqrt{s}\|\boldsymbol{f}\|_2\}$. We use this inequality to replace $\|\boldsymbol{h}\|_1$ term in equation 18 and we get

$$\left(\frac{\sqrt{C_{\min}(\boldsymbol{\Sigma})}}{4} - 36\sqrt{\frac{\xi(\boldsymbol{\Sigma})k \log p}{N}}\right)\|\boldsymbol{h}\|_2 - 27\lambda\sqrt{\frac{\xi(\boldsymbol{\Sigma})s \log p}{N}}\|\boldsymbol{f}\|_2 \leq \frac{1}{\sqrt{N}}\|\boldsymbol{Ah}\|_2, \quad (19)$$

or

$$\left(\frac{\sqrt{C_{\min}(\boldsymbol{\Sigma})}}{4} - 36\sqrt{\frac{\xi(\boldsymbol{\Sigma})k \log p}{N}}\right)\|\boldsymbol{h}\|_2 + \left(\sqrt{\frac{n}{N}} - 27\lambda\sqrt{\frac{\xi(\boldsymbol{\Sigma})s \log p}{N}}\right)\|\boldsymbol{f}\|_2 \leq \frac{1}{\sqrt{N}}\|\boldsymbol{Ah}\|_2 + \sqrt{\frac{n}{N}}\|\boldsymbol{f}\|_2. \quad (20)$$

The assumption $N \geq c_2 \frac{\xi(\boldsymbol{\Sigma})}{C_{\min}(\boldsymbol{\Sigma})} k \log p$ in the lemma implies $36\sqrt{\frac{\xi(\boldsymbol{\Sigma}) k \log p}{N}} \leq \frac{\sqrt{C_{\min}(\boldsymbol{\Sigma})}}{8}$ for $c_2 \geq 288^2$. Similarly, the assumption $s \leq c_1 \frac{N}{\rho^2 \log n}$ in the lemma implies $27\lambda\sqrt{\frac{\xi(\boldsymbol{\Sigma}) s \log p}{N}} \leq \frac{1}{2}\sqrt{\frac{n}{N}}$ for $c_1 \leq 1/54^2$. Using these two inequalities, the two terms inside the brackets can be simplified to obtain the following:

$$\frac{\sqrt{C_{\min}(\boldsymbol{\Sigma})}}{8}\|\boldsymbol{h}\|_2 + \frac{1}{2}\sqrt{\frac{n}{N}}\|\boldsymbol{f}\|_2 \leq \frac{1}{\sqrt{N}}\|\boldsymbol{Ah}\|_2 + \sqrt{\frac{n}{N}}\|\boldsymbol{f}\|_2. \tag{21}$$

After a straightforward application of Lemma 5, we further get

$$\frac{C_{\min}(\boldsymbol{\Sigma})}{128}\|\boldsymbol{h}\|_2^2 + \frac{n}{8N}\|\boldsymbol{f}\|_2^2 \leq \frac{1}{N}\|\boldsymbol{Ah}\|_2^2 + \frac{n}{N}\|\boldsymbol{f}\|_2^2,$$

or

$$\min\left(\frac{C_{\min}(\boldsymbol{\Sigma})}{128}, \frac{n}{8N}\right)(\|\boldsymbol{h}\|_2^2 + \|\boldsymbol{f}\|_2^2) \leq \frac{1}{N}\|\boldsymbol{Ah}\|_2^2 + \frac{n}{N}\|\boldsymbol{f}\|_2^2, \tag{22}$$

which completes the proof. $\qquad\square$

**Lemma 2** (**Mutual incoherence condition (Nguyen & Tran, 2012)**). *Consider the Gaussian sensing matrix $\boldsymbol{A}_2 \in \mathbb{R}^{n \times p}$ whose rows are i.i.d. $\mathcal{N}(\boldsymbol{0}, \boldsymbol{\Sigma})$. We have the set $\mathcal{C} = \{(\boldsymbol{h}, \boldsymbol{f}) \in (\mathbb{R}^p \times \mathbb{R}^n)$ such that $\|\boldsymbol{h}_{\boldsymbol{T}^c}\|_1 + \lambda\|\boldsymbol{f}_{\boldsymbol{S}^c}\|_1 \leq 3\|\boldsymbol{h}_{\boldsymbol{T}}\|_1 + 3\lambda\|\boldsymbol{f}_{\boldsymbol{S}}\|_1\}$ with $|T| = k$ and $|S| = s$ as defined earlier. Select $\lambda = \rho\sqrt{\frac{n}{N}\frac{\log n}{\xi(\boldsymbol{\Sigma}) \log p}}$, where $\rho \in (0, 1)$ is a constant. Assume that $s \leq \min\left(\frac{N}{n}\frac{\xi(\boldsymbol{\Sigma})}{C_{min}(\boldsymbol{\Sigma})}\frac{k \log p}{\rho^2 \log n}, c_5 \frac{C_{min}(\boldsymbol{\Sigma})}{C_{max}(\boldsymbol{\Sigma})}n\right)$ and $n \geq c_6\xi(\boldsymbol{\Sigma})\frac{C_{max}(\boldsymbol{\Sigma})}{C_{min}^2(\boldsymbol{\Sigma})}k \log p$ for some sufficiently small positive constant $c_5$ and sufficiently large constant $c_6$, then the following inequality holds with probability greater than $1 - \exp(-c_7 n)$:*

$$\frac{1}{\sqrt{n}}|\langle \boldsymbol{A}_2\boldsymbol{h}, \boldsymbol{f}\rangle| \leq k_m(\|\boldsymbol{h}\|_2 + \|\boldsymbol{f}\|_2)^2 \; \forall \; (\boldsymbol{h}, \boldsymbol{f}) \in \mathcal{C}, \tag{23}$$

*where $c_7, k_m$ are positive constants. We refer to $k_m$ as the mutual incoherence constant.*

**Lemma 3.** *Consider the Gaussian sensing matrix $\boldsymbol{A}_1 \in \mathbb{R}^{m \times p}$ and $\boldsymbol{A}_2 \in \mathbb{R}^{n \times p}$ whose rows are i.i.d. $\mathcal{N}(\boldsymbol{0}, \boldsymbol{\Sigma})$. We have the set $\mathcal{C} = \{(\boldsymbol{h}, \boldsymbol{f}) \in (\mathbb{R}^p \times \mathbb{R}^n)$ such that $\|\boldsymbol{h}_{\boldsymbol{T}^c}\|_1 + \lambda\|\boldsymbol{f}_{\boldsymbol{S}^c}\|_1 \leq 3\|\boldsymbol{h}_{\boldsymbol{T}}\|_1 + 3\lambda\|\boldsymbol{f}_{\boldsymbol{S}}\|_1\}$ with $|T| = k$ and $|S| = s$ as defined earlier. Select $\lambda = \rho\sqrt{\frac{n}{N}\frac{\log n}{\xi(\boldsymbol{\Sigma}) \log p}}$, where $\rho \in (0, 1)$ is a constant. Assume that $s \leq \min\left(c_1 \frac{N}{\rho^2 \log n}, \frac{N}{n}\frac{\xi(\boldsymbol{\Sigma})}{C_{min}(\boldsymbol{\Sigma})}\frac{k \log p}{\rho^2 \log n}, c_5 \frac{C_{min}(\boldsymbol{\Sigma})}{C_{max}(\boldsymbol{\Sigma})}n\right)$, $N \geq c_2 \frac{\xi(\boldsymbol{\Sigma})}{C_{min}(\boldsymbol{\Sigma})}k \log p$, $n \geq c_6\xi(\boldsymbol{\Sigma})\frac{C_{max}(\boldsymbol{\Sigma})}{C_{min}^2(\boldsymbol{\Sigma})}k \log p$ and $\boldsymbol{A}_2$ satisfies the mutual incoherence condition stated in Lemma 2 with mutual incoherence constant $k_m < \min\left(\frac{N}{n}\frac{C_{min}(\boldsymbol{\Sigma})}{512}, \frac{1}{32}\right)$. Then the following inequality holds with probability greater than $1 - c_3 \exp(-c_4 N) - \exp(-c_7 n)$:*

$$\min\left(\left(\frac{C_{\min}(\boldsymbol{\Sigma})}{512}, \frac{n}{32N}\right) - k_m\frac{n}{N}\right)(\|\boldsymbol{h}\|_2 + \|\boldsymbol{f}\|_2)^2 \leq \frac{1}{2N}\|\boldsymbol{A}_1\boldsymbol{h}\|_2^2 + \frac{1}{2N}\|\boldsymbol{A}_2\boldsymbol{h} + \sqrt{n}\boldsymbol{f}\|_2^2 \; \forall \; (\boldsymbol{h}, \boldsymbol{f}) \in \mathcal{C}, \tag{24}$$

*where $c_1, c_2, c_3, c_4, c_5, c_6, c_7$ are positive constants.*

*Proof.* The proof involves applying Lemma 1 and Lemma 2 to lower bound the term $\frac{1}{2N}\|\boldsymbol{A}_1\boldsymbol{h}\|_2^2 + \frac{1}{2N}\|\boldsymbol{A}_2\boldsymbol{h} + \sqrt{n}\boldsymbol{f}\|_2^2$. To start, we have

$$
\begin{aligned}
\frac{1}{2N}\|\boldsymbol{A}_1\boldsymbol{h}\|_2^2 + \frac{1}{2N}\|\boldsymbol{A}_2\boldsymbol{h} + \sqrt{n}\boldsymbol{f}\|_2^2 &= \frac{1}{2N}\|\boldsymbol{A}_1\boldsymbol{h}\|_2^2 + \frac{1}{2N}\|\boldsymbol{A}_2\boldsymbol{h}\|_2^2 + \frac{n}{2N}\|\boldsymbol{f}\|_2^2 + \frac{\sqrt{n}}{N}\langle \boldsymbol{A}_2\boldsymbol{h}, \boldsymbol{f}\rangle \\
&\geq \frac{1}{2N}\|\boldsymbol{A}_1\boldsymbol{h}\|_2^2 + \frac{1}{2N}\|\boldsymbol{A}_2\boldsymbol{h}\|_2^2 + \frac{n}{2N}\|\boldsymbol{f}\|_2^2 - \frac{\sqrt{n}}{N}|\langle \boldsymbol{A}_2\boldsymbol{h}, \boldsymbol{f}\rangle| \\
&= \frac{1}{2N}\|\boldsymbol{Ah}\|_2^2 + \frac{n}{2N}\|\boldsymbol{f}\|_2^2 - \frac{\sqrt{n}}{N}|\langle \boldsymbol{A}_2\boldsymbol{h}, \boldsymbol{f}\rangle| \\
&\geq \min\left(\frac{C_{\min}(\boldsymbol{\Sigma})}{256}, \frac{n}{16N}\right)(\|\boldsymbol{h}\|_2^2 + \|\boldsymbol{f}\|_2^2) - k_m\frac{n}{N}(\|\boldsymbol{h}\|_2 + \|\boldsymbol{f}\|_2)^2.
\end{aligned}
\tag{25}
$$

We used Lemma 1 and Lemma 2 in the last step. Next, we use the fact that $(a^2 + b^2) \geq \frac{1}{2}(a+b)^2 \ \forall a, b > 0$ to obtain

$$
\begin{aligned}
\frac{1}{2N}\|\boldsymbol{A_1 h}\|_2^2 + \frac{1}{2N}\|\boldsymbol{A_2 h} + \sqrt{n}\boldsymbol{f}\|_2^2 &\geq \min\left(\frac{C_{\min}(\boldsymbol{\Sigma})}{512}, \frac{n}{32N}\right)(\|\boldsymbol{h}\|_2 + \|\boldsymbol{f}\|_2)^2 - k_m\frac{n}{N}(\|\boldsymbol{h}\|_2 + \|\boldsymbol{f}\|_2)^2 \\
&= \left(\min\left(\frac{C_{\min}(\boldsymbol{\Sigma})}{512}, \frac{n}{32N}\right) - k_m\frac{n}{N}\right)(\|\boldsymbol{h}\|_2 + \|\boldsymbol{f}\|_2)^2,
\end{aligned}
\tag{26}
$$

which completes the proof.

$\square$

**Lemma 4.** *Consider the Gaussian sensing matrices $\boldsymbol{A}_1 \in \mathbb{R}^{m \times p}$ and $\boldsymbol{A}_2 \in \mathbb{R}^{n \times p}$ with i.i.d. $\mathcal{N}(0, 1/N)$ entries. There exist positive constants $c_8, c_9$ such that the augmented matrix $\boldsymbol{H} = \begin{bmatrix} \boldsymbol{A}_1 & \boldsymbol{0}_{m \times n} \\ \boldsymbol{A}_2 & \boldsymbol{I}_{n \times n} \end{bmatrix}$ satisfies the structured-sparsity restricted isometry property (SS-RIP) of order $[(p, k), (n, s)]$ provided that $k \log(p/k) + s \log(n/s) \leq c_8 N$, with probability atleast $1 - 3\exp(-c_9 N)$. The constants $c_8, c_9$ depend on the restricted isometry constant $\delta$. Equivalently, we have the following result:*

$$
\mathbb{P}\left((1-\delta)\|\boldsymbol{x}\|_2^2 \leq \|\boldsymbol{H}\boldsymbol{x}\|_2^2 \leq (1+\delta)\|\boldsymbol{x}\|_2^2 \text{ for all } \boldsymbol{x} \text{ such that } \|\boldsymbol{x}(1:p)\|_0 \leq k \text{ and } \|\boldsymbol{x}(p+1:p+n)\|_0 \leq s\right) \geq
$$
$$
1 - 3\exp(-c_9 N). \tag{27}
$$

*Proof.* Referring to the notation in the main paper, we expand the term $\|\boldsymbol{H}\boldsymbol{x}\|_2^2$ as following:

$$
\begin{aligned}
\|\boldsymbol{H}\boldsymbol{x}\|_2^2 &= \|\boldsymbol{A}_1\boldsymbol{\beta}\|_2^2 + \|\boldsymbol{A}_2\boldsymbol{\beta} + \boldsymbol{z}\|_2^2 \\
&= \|\boldsymbol{A}_1\boldsymbol{\beta}\|_2^2 + \|\boldsymbol{A}_2\boldsymbol{\beta}\|_2^2 + \|\boldsymbol{z}\|_2^2 + 2\boldsymbol{z}^T\boldsymbol{A}_2\boldsymbol{\beta} \\
&= \|\boldsymbol{A}\boldsymbol{\beta}\|_2^2 + \|\boldsymbol{z}\|_2^2 + 2\boldsymbol{z}^T\boldsymbol{A}_2\boldsymbol{\beta}.
\end{aligned}
\tag{28}
$$

Since $\boldsymbol{A}_2$ has i.i.d. $\mathcal{N}(0, 1/N)$ entries, we have that $2\boldsymbol{z}^T\boldsymbol{A}_2\boldsymbol{\beta} \sim \mathcal{N}(0, \frac{4}{N}\|\boldsymbol{\beta}\|_2^2\|\boldsymbol{z}\|_2^2)$. Consequently, $\mathbb{P}(|2\boldsymbol{z}^T\boldsymbol{A}_2\boldsymbol{\beta}| \geq \epsilon_1\|\boldsymbol{\beta}\|_2\|\boldsymbol{z}\|_2) = \mathbb{P}\left(\frac{|2\boldsymbol{z}^T\boldsymbol{A}_2\boldsymbol{\beta}|}{\frac{2\|\boldsymbol{\beta}\|_2\|\boldsymbol{z}\|_2}{\sqrt{N}}} \geq \epsilon_1\frac{\sqrt{N}}{2}\right) = 2Q\left(\epsilon_1\frac{\sqrt{N}}{2}\right)$ where $Q(.)$ denotes the tail integral of the standard normal distribution. Using the result that $Q(t) \leq \frac{1}{2}\exp(-t^2/2)$, we obtain

$$
\mathbb{P}\left(|2\boldsymbol{z}^T\boldsymbol{A}_2\boldsymbol{\beta}| \leq \epsilon_1\|\boldsymbol{\beta}\|_2\|\boldsymbol{z}\|_2\right) \geq 1 - \exp(-N\epsilon_1^2/8). \tag{29}
$$

Next, using the Gaussian concentration results, we have that

$$
\mathbb{P}\left((1-\epsilon_2)\|\boldsymbol{\beta}\|_2^2 \leq \|\boldsymbol{A}\boldsymbol{\beta}\|_2^2 \leq (1+\epsilon_2)\|\boldsymbol{\beta}\|_2^2\right) \geq 1 - 2\exp(-N\epsilon_2^2/8). \tag{30}
$$

Next, applying intersection bound with equation 29 and equation 30, we get

$$
(1-\epsilon_2)\|\boldsymbol{\beta}\|_2^2 - \epsilon_1\|\boldsymbol{\beta}\|_2\|\boldsymbol{z}\|_2 \leq \|\boldsymbol{A}\boldsymbol{\beta}\|_2^2 + 2\boldsymbol{z}^T\boldsymbol{A}_2\boldsymbol{\beta} \leq (1+\epsilon_2)\|\boldsymbol{\beta}\|_2^2 + \epsilon_1\|\boldsymbol{\beta}\|_2\|\boldsymbol{z}\|_2 \text{ w.p. atleast}
$$
$$
1 - \exp(-N\epsilon_1^2/8) - 2\exp(-N\epsilon_2^2/8). \tag{31}
$$

Adding $\|\boldsymbol{z}\|_2^2$ to equation 31, we obtain

$$
(1-\epsilon_2)\|\boldsymbol{\beta}\|_2^2 - \epsilon_1\|\boldsymbol{\beta}\|_2\|\boldsymbol{z}\|_2 + \|\boldsymbol{z}\|_2^2 \leq \|\boldsymbol{H}\boldsymbol{x}\|_2^2 \leq (1+\epsilon_2)\|\boldsymbol{\beta}\|_2^2 + \epsilon_1\|\boldsymbol{\beta}\|_2\|\boldsymbol{z}\|_2 + \|\boldsymbol{z}\|_2^2 \text{ w.p. atleast}
$$
$$
1 - \exp(-N\epsilon_1^2/8) - 2\exp(-N\epsilon_2^2/8). \tag{32}
$$

Denote $\epsilon = \epsilon_1 + \epsilon_2$. Next, using the results $\|\boldsymbol{\beta}\|_2^2 + \|\boldsymbol{z}\|_2^2 = \|\boldsymbol{x}\|_2^2$, $\|\boldsymbol{\beta}\|_2\|\boldsymbol{z}\|_2 \leq \|\boldsymbol{x}\|_2^2$ and $\|\boldsymbol{\beta}\|_2^2 \leq \|\boldsymbol{x}\|_2^2$, equation 32 can be simplified to obtain the following inequality:

$$
(1 - \epsilon)\|\boldsymbol{x}\|_2^2 \leq \|\boldsymbol{H}\boldsymbol{x}\|_2^2 \leq (1 + \epsilon)\|\boldsymbol{x}\|_2^2 \text{ w.p. atleast } 1 - \exp(-N\epsilon_1^2/8) - 2\exp(-N\epsilon_2^2/8). \tag{33}
$$

The aforementioned inequality is satisfied for any $\boldsymbol{x}$ with the specified probability. However, we have some structure in the sparsity of $\boldsymbol{x}$, that is, $\|\boldsymbol{x}(1:p)\|_0 \leq k$ and $\|\boldsymbol{x}(p+1:p+n)\|_0 \leq s$. And hence, we restrict our attention to such $(k+s)$ sparse $\boldsymbol{x}$. Let $J$ denote a set whose elements are all such $\binom{p}{k}\binom{n}{s}$ possible support sets. We denote the individual elements in $J$ by $J_i$, where $i = 1, 2, 3, \ldots, \binom{p}{k}\binom{n}{s}$. Using Lemma 5.1 in Baraniuk et al. (2008), we have the following result.: For any $J_i \in J$ and any $\delta \in (0,1)$, the following inequality is satisfied with probability atleast $1 - (12/\delta)^{(k+s)}(\exp(-N\delta^2/128) + 2\exp(-N\delta^2/128))$ or $1 - 3(12/\delta)^{(k+s)}\exp(-N\delta^2/128)$:

$$(1+\delta)\|\boldsymbol{x}\|_2^2 \leq \|\boldsymbol{H}\boldsymbol{x}\|_2^2 \leq (1+\delta)\|\boldsymbol{x}\|_2^2 \text{ for all } \boldsymbol{x} \in \mathbb{R}^{p+n} \text{ with support } J_i. \tag{34}$$

Note that, within Lemma 5.1 in Baraniuk et al. (2008), we chose $\epsilon_1 = \epsilon_2 = \delta/4$ and accordingly $\epsilon = \delta/2$.

We denote the event

$$E_i := (1+\delta)\|\boldsymbol{x}\|_2^2 \leq \|\boldsymbol{H}\boldsymbol{x}\|_2^2 \leq (1+\delta)\|\boldsymbol{x}\|_2^2 \text{ for all } \boldsymbol{x} \in \mathbb{R}^{p+n} \text{ with support } J_i, \tag{35}$$

where $i = 1, 2, 3, \ldots, \binom{p}{k}\binom{n}{s}$. From equation 34, we have that

$$\mathbb{P}(\bar{E}_i) \leq 3(12/\delta)^{(k+s)}\exp(-N\delta^2/128). \tag{36}$$

Using equation 36 with union bound, we have that

$$\mathbb{P}\left(\cup_{i=1}^{\binom{p}{k}\binom{n}{s}} \bar{E}_i\right) \leq \sum_{i=1}^{\binom{p}{k}\binom{n}{s}} \mathbb{P}(\bar{E}_i) \leq 3\binom{p}{k}\binom{n}{s}(12/\delta)^{(k+s)}\exp(-N\delta^2/128), \tag{37}$$

or

$$\mathbb{P}\left(\cap_{i=1}^{\binom{p}{k}\binom{n}{s}} E_i\right) \geq 1 - 3\binom{p}{k}\binom{n}{s}(12/\delta)^{(k+s)}\exp(-N\delta^2/128), \tag{38}$$

or

$$\mathbb{P}\left((1-\delta)\|\boldsymbol{x}\|_2^2 \leq \|\boldsymbol{H}\boldsymbol{x}\|_2^2 \leq (1+\delta)\|\boldsymbol{x}\|_2^2 \text{ for all } \boldsymbol{x} \text{ with } \|\boldsymbol{x}(1:p)\|_0 \leq k \text{ and } \|\boldsymbol{x}(p+1:p+n)\|_0 \leq s\right) \geq$$
$$1 - 3\binom{p}{k}\binom{n}{s}(12/\delta)^{(k+s)}\exp(-N\delta^2/128). \tag{39}$$

Now, it only remains to simplify the term on the right-hand side. Assume that $k\log(p/k) + s\log(n/s) \leq c_8 N$ for some $c_8 > 0$. With this assumption and using the well-known results that $\binom{p}{k} \leq (ep/k)^k$ and $\binom{n}{s} \leq (en/s)^s$, the term $3\binom{p}{k}\binom{n}{s}(12/\delta)^{(k+s)}\exp(-N\delta^2/128)$ in equation 39 can be upper-bounded as following:

$$3\binom{p}{k}\binom{n}{s}(12/\delta)^{(k+s)}\exp(-N\delta^2/128) \leq 3\left(\frac{ep}{k}\right)^k\left(\frac{en}{s}\right)^s\left(\frac{12}{\delta}\right)^{(k+s)}\exp(-N\delta^2/128),$$
$$\leq 3\exp\left((k+s)(1+\log(12/\delta)) + c_8 N - \frac{N\delta^2}{128}\right),$$
$$\leq 3\exp\left(\left[(1+\log(12/\delta))\left(\frac{1}{\log(p/k)} + \frac{1}{\log(n/s)}\right) + 1\right]c_8 N - \frac{N\delta^2}{128}\right), \tag{40}$$

where we use $k\log(p/k) + s\log(n/s) \leq c_8 N$ in the second last step and $k \leq c_8 N/\log(p/k)$ and $s \leq c_8 N/\log(n/s)$ in the last step. Denote $c_9 := -\left[(1+\log(12/\delta))\left(\frac{1}{\log(p/k)} + \frac{1}{\log(n/s)}\right) + 1\right]c_8 + \frac{\delta^2}{128}$. We can always choose $c_8 > 0$ sufficiently small to ensure that $c_9 > 0$. Consequently, we have that

$$3\binom{p}{k}\binom{n}{s}(12/\delta)^{(k+s)}\exp(-N\delta^2/128) \leq 3\exp(-c_9 N). \tag{41}$$

| Ar-Lasso with ZSC | R-Lasso with ZSC | $\ell_1$-Htp | $\ell_2$-Htp |
|---|---|---|---|
| 0.116 secs | 0.151 secs | 3.327 secs | 0.533 secs |

Table 1: Time taken for the four unlabeled sensing algorithms to execute

From equation 39 and equation 41, we get the following final result:

$$\mathbb{P}\left((1-\delta)\|\boldsymbol{x}\|_2^2 \leq \|\boldsymbol{Hx}\|_2^2 \leq (1+\delta)\|\boldsymbol{x}\|_2^2 \text{ for all } \boldsymbol{x} \text{ with } \|\boldsymbol{x}(1:p)\|_0 \leq k \text{ and } \|\boldsymbol{x}(p+1:p+n)\|_0 \leq s\right) \geq$$
$$1 - 3\exp\left(-c_9 N\right). \quad (42)$$

$\square$

**Lemma 5.** *Let $a, b, c, d \geq 0$ and assume that $c + d \leq a + b$. Then, we have that:*

$$\frac{1}{2}(c^2 + d^2) \leq a^2 + b^2. \quad (43)$$

*Proof.* The above result is a trivial application of Cauchy-Schwarz inequality. Take $\boldsymbol{u} = (1,1)$ and $\boldsymbol{v} = (a, b)$. Then we have that $|\langle \boldsymbol{u}, \boldsymbol{v} \rangle|^2 \leq \|\boldsymbol{u}\|_2^2 \|\boldsymbol{v}\|_2^2$ or $(a+b)^2 \leq 2(a^2+b^2)$ or $(c+d)^2 \leq 2(a^2+b^2)$ or $\frac{1}{2}(c^2+d^2) \leq (a^2+b^2)$. $\square$

## 3 Experiments with Execution Timings

In Table 1, we show the time taken for the four algorithms to execute for $p = 240$, $N = 120$, $k = 14$, $m = 32$, $s = 16$ and 2% measurement noise. The timing values in Table 1 are averaged over 50 noise and permutation instances. These timing values do not include the time taken for choosing the best hyper-parameters using cross-validation for any of the methods. Note that Ar-Lasso is around five times faster than $\ell_2$-Htp. $\ell_1$-Htp is the slowest among them as it requires a larger number of iteration to finish and also because of the computationally expensive $\ell_1$-norm optimization. In summary, Ar-Lasso is more efficient, timing-wise while $\ell_2$-Htp estimates $\boldsymbol{\beta}^*$ more accurately.