# OpenReview forum: "Unlabelled Compressive Sensing under Sparse Permutation and Prior Information"
_TMLR — Accepted by TMLR_

### Review · Reviewer_9Gpt · 2024-11-27

**Summary Of Contributions:**

The paper considers the problem of unlabelled compressed sensing, where the correspondences between the measurements and the rows of the sensing matrix are unknown, and the number of measurements is smaller than the variable dimension. While the problem is difficult, it is made tractable via a few assumptions. The assumptions are that the ground truth vector is sparse and that some correspondences are given. Under these assumptions, the paper makes several contributions:
- it formulates a LASSO-type robust recovery problem and provides recovery guarantees (Theorem 1);
- it proposes an algorithm based on hard-thresholding pursuit and proves its convergence (Theorem 2);
- it has experimental validations on synthetic data and for image registration (in the appendix).

**Audience:**

Yes

**Broader Impact Concerns:**

I do not see any concerns on the ethical implications of the work.

**Claims And Evidence:**

Yes

**Requested Changes:**

Besides the baseline mentioned above, here are a few suggestions on style and technical presentation:

Some citations hurt the readability of the corresponding sentences. For example, in the first page, it was written:
> *Unnikrishnan et al. Unnikrishnan et al. (2018) and Han et al. Han et al. (2018) state that ...*

This is not readable, as the author names are repeated. I would recommend using \citet and \citep rather than \cite, and use the common practice for in-text citations, see, e.g.,:
> https://tex.stackexchange.com/questions/480040/citation-for-an-addendum-in-brackets-citet-vs-citep
> https://blog.apastyle.org/apastyle/2011/01/writing-in-text-citations-in-apa-style.html

Similarly,

- The definition of set $C$ at the beginning of Section 3.1 should have an independent equation number, so that in latter lemmas it can be referenced. Currently, the definitions are repeating.
- To make the paper more readable and friendly to novice readers, the definition of a mathematical notation should be followed by a sentence that explains it in words. This is currently missing in many definitions:
    - the set $C$
    - the GEREC condition (it should have an independent equation number)
    - the mutual coherence condition (it should have an independent equation number)
- Please unify the use of *generalized* (*generalization*) and *generalised* (*generalisation).
- The first sentence of Theorem 1, shown below is not readable, as it writes Eq. 5 rather than (5), and it is not comfortable to read * the optimization problem Eq. 5*
> *Consider the optimization problem Eq. 5 and the observation model Eq. 1 under the assumptions (C1)-(C4).*
- In Line 4 of Algorithm 1, $k$ and $s$ are not in math fonts. Also, in many places, $\max$ and $\min$ are not in math fonts.
- The explanation of Algorithm 1 could be made better. Specifically, the texts should explain the algorithm line by line, rather than step by step (the step number in use does not match the line number).
- In Definition 1, the phrases *for every $(k+s)$-sparse $g$* could be removed as the latter conditions on its $\ell\_0$ norm makes sure that $g$ is $(k+s)$-sparse
- In equations (8) and (9), the commas are unnecessary and they actually interrupt the flow.
- Equation (11) should be rearranged as it is too long.
- The name *RRMSE* is weird, as MSE typically represents *mean squared error*, while the actual definition is not the $\ell\_2$ norm squared.

What is $w$ in equation (10)? Did I miss its definition? Some discussion on the error term of equation (10) is needed, as the algorithm might not converge if the error term is too large.

**Strengths And Weaknesses:**

Strength:
- The paper is easy to follow, and the writing is clear;
- Related works are clearly explained. E.g., the comparison and relation to prior work (Nguyen & Tran 2012) are discussed in detail;
- The paper has solid contributions that cover theory, algorithms, and applications.

Weaknesses:
- While the paper provides wide coverage, the technical contributions are a little bit incremental, as the results appear to be natural extensions of prior works.
- The style and technical presentation could be improved (see "Requested Changes").
- Compared to prior works, and my knowledge, the paper introduces an unconventional assumption (C3), which requires $m$ correspondences given. While the justification is that in some image applications, there could be a few hand-calibrated correspondences available, I personally find this assumption needs more interrogations from both theoretical and empirical perspectives. Specifically,  consider the formulation/algorithm: minimize the $\ell\_1$ norm of $\beta$ subject to the constraints $A\_1 \beta=y_1$ (or formulate it as a LASSO problem). The rationale is that, if $m$ is large enough, then we could potentially recover $\beta^*$ just from the $m$ correspondences (and if so, we could ignore the potentially permuted measurements). This leads to two questions:
    - In theory, I would assume recovery guarantees would exist for this baseline. The paper should compare Theorem 1 with this guarantee, as this would give us an idea about the usefulness of learning from permuted measurements. We would expect that Theorem 1 would indeed be better, as the extreme case $m=0$ suggests this baseline would fail.
    - For experiments, this baseline could have been included in the comparison. This is a different one, compared to R-LASSO.

Questions:
- is there a typo in the definition of the R-LASSO baseline? That is, should $e$ be an optimization variable as well? Should it be $A\_2$ and $y\_2$?

---

> ### Author Response · Authors · 2024-12-17
> **Response to reviewer 9Gpt: Part 1**
>
> We thank the reviewer for their detailed review and are responding below to most comments. Responses to the remaining comments will be posted separately due to space limitations here.
>
> 1) *The paper provides wide coverage, the technical contributions are a bit incremental, as the results appear to be natural extensions of prior works.*
>
> There are very few papers on unlabeled compressed sensing. The one paper (Liangzu Peng, Boshi Wang, and Manolis Tsakiris. Homomorphic sensing: Sparsity and noise. In Int. Conf. Mach. Learn., volume 139 of Proc. Mach. Learn. Res., pp. 8464–8475. PMLR, 2021.) that treats this topic presents theoretical analysis, but the analysis in that paper does not apply to any practical or computationally tractable estimator. Even though the techniques we have presented are simple, our paper is the first one to present theoretical performance bounds for a tractable estimator for the problem of unlabeled compressed sensing.
>
> 2) *The paper introduces an unconventional assumption (C3), which requires $m$ correspondences given. While the justification is that in some image applications, there could be a few hand-calibrated correspondences available, I personally find this assumption needs more interrogations from both theoretical and empirical perspectives.*
>
> In image alignment applications in varied domains including medical imaging, it is quite common to have experts mark a *small* number of pairs of corresponding points in the two images. These act as the $m$ accurate correspondences given as input. We therefore believe this is not a contrived assumption, and is in tune with this application. Moreover, the estimators we have presented as well as their theoretical bounds also easily extend to the case when $m = 0$.
>
> 3) *Consider the formulation: minimize the $\ell_1$  norm of $\boldsymbol{\beta}$
>  subject to $\boldsymbol{y} = \boldsymbol{Ax}$  (or formulate it as a LASSO problem). The rationale is that, if $m$
>  is large enough, then we could potentially recover  just from the $m$ correspondences (and if so, we could ignore the potentially permuted measurements). This leads to two questions: In theory, I would assume recovery guarantees would exist for this baseline. The paper should compare Theorem 1 with this guarantee, as this would give us an idea about the usefulness of learning from permuted measurements. We would expect that Theorem 1 would indeed be better, as the extreme case $m=0$ suggests this baseline would fail.*
>
> If $m$ is large enough so that the matrix $\boldsymbol{A_2}$ obeys the restricted eigenvalue condition, then we may not even require the permuted measurements at all. In such a case, the problem just reduces to a simple LASSO and the error bound would be as follows:
> $$\|\boldsymbol{\tilde{\beta}}-\boldsymbol{\beta^*}\|_2 \leq O(\sigma k \sqrt{\log p/ m})$$
>
> We agree with the reviewer that this is a useful baseline (see our response to the next point as well). However such a large number of correspondences will generally not be available in applications. Sensing matrices randomly generated by Rademacher or Gaussian distributions would need to contain at least $O(k \log p)$ rows in order to recover $k$-sparse vectors in $\mathbb{R}^p$ successfully. If $m$ is smaller than that, then successful recovery will not be achievable, and hence using the sparsely permuted measurements will be inevitable.
>
> 4) *For experiments, this baseline could have been included in the comparison. This is a different one, compared to R-LASSO.*
>
> We have already done this comparison, but we will mention it in the revised version explicitly. When $m$ is small, the reconstruction errors are too high compared to any of the other methods. Also, as mentioned in the previous point, a large value of $m$ is not practical.
>
> 5) *Is there a typo in the definition of the R-LASSO baseline? Should $\boldsymbol{e}$  be an optimization variable as well?*
>
> The reviewer is right. In the definition of R-LASSO, both $\boldsymbol{\beta}$ and $\boldsymbol{e}$ are optimization variables. We will correct this in the revision. However, this estimator does not assume any known correspondences due to which $\boldsymbol{y_1}$ is an empty vector and $\boldsymbol{A_1}$ is an empty matrix. Hence for this estimator $\boldsymbol{A} = \boldsymbol{A_2}$ and $\boldsymbol{y} = \boldsymbol{y}_2$ where $\boldsymbol{y}_2$ and $\boldsymbol{A}_2$ are defined in the main paper in the beginning of Section 2.
>
> 6) *What is $\boldsymbol{w}$ in equation (10)? Some discussion on the error term of (10) is needed, as the algorithm might not converge if it is too large.*
>
> The term $\boldsymbol{w}$ refers to the measurement noise vector from equation (1). In the revised version, we will state this explicitly in Theorem 2 for clarity. Error terms involving the noise vector are very typically in the analysis of pursuit algorithms like OMP and others, and it does not depend on the signal.

---

> > ### Author Response · Authors · 2024-12-17
> > **Response to reviewer 9Gpt: Part 2**
> >
> > 7) *I recommend using citet and citep rather than cite.*
> >
> > The reviewer is right. We will correct this in the revised version.
> >
> > 8) *The definition of set $C$  at the beginning of Section 3.1 should have an independent equation number, so that in latter lemmas it can be referenced. Currently, the definitions are repeating.*
> >
> > The reviewer is right. We will correct this in the revised version.
> >
> > 9) *To make the paper more readable and friendly to novice readers, the definition of a mathematical notation should be followed by a sentence that explains it in words. This is currently missing in many definitions: the set $C$; the GEREC condition (it should have an independent equation number); the mutual coherence condition (it should have an independent equation number).*
> >
> > We agree with the reviewer that doing so will improve the readability of the paper. We will make these changes in the revised version. We mention the additions here:
> >
> > a) The set $\mathcal{C}$ is a natural restriction on the error vectors $\boldsymbol{h} := \boldsymbol{\tilde{\beta}} - \boldsymbol{\beta^*}$ and $e := \boldsymbol{\tilde{\delta}} - \boldsymbol{\delta^*}$ that emerges from this optimization problem. This can be seen from the proof of Theorem 1 in the supplemental material, starting from step (S3) through to (S13). We will improve the description of some of the steps in the supplemental material.
> >
> > b) The GEREC is essentially stating that the sum of the squared magnitudes of the vectors $\boldsymbol{Ah}$ and $\boldsymbol{f}$ is lower-bounded by a constant factor times $\|\boldsymbol{h}\|^2_2 + \|\boldsymbol{f}\|^2_2$ for any $(\boldsymbol{h},\boldsymbol{f}) \in \mathcal{C}$. This constant factor depends on the sensing matrix, and very small constant factors are not desirable, i.e. "good" sensing matrices will have larger constant factors.
> >
> > c) The mutual coherence condition states that the column vectors of $\boldsymbol{A}$ are sufficiently dissimilar compared to columns of the identity matrix. This ensures that there is clear separation between the signal vector $\boldsymbol{\beta}$ and the sparse error vector $\boldsymbol{e}$.
> >
> > 10) *Please unify the use of generalized (generalization) and generalised (generalisation).*
> >
> > We agree and we will maintain consistency in the revised version.
> >
> > 11) *The first sentence of Theorem 1, shown below is not readable, as it writes Eq. 5 rather than (5), and it is not comfortable to read the optimization problem Eq. 5  in Consider the optimization problem Eq. 5 and the observation model Eq. 1 under the assumptions (C1)-(C4).*
> >
> > We agree and we will restate the first sentence of Theorem 1 as follows:
> > ``Consider the optimization problem in (5) and the observation model in (1)...''. We will use eqref throughout instead of Eqn. followed by a number.
> >
> > 12) *In Line 4 of Algorithm 1, $k$ and $s$  are not in math fonts. Also, in many places, min and max are not in math fonts.*
> >
> > We agree and we will fix this in the revision.
> >
> > 13) * The explanation of Algorithm 1 could be made better. Specifically, the texts should explain the algorithm line by line, rather than step by step (the step number in use does not match the line number).*
> >
> >  We agree and we will fix this in the revised version.
> >
> > 14) *In Definition 1, the phrase for every $(k+s)$-sparse $\boldsymbol{g}$ could be removed as the latter conditions on its $\ell_0$ norm makes sure that it is $(k+s)$-sparse.*
> >
> > We agree and we will fix this in the revised version.
> >
> > 15)  *In equations (8) and (9), the commas are unnecessary and they actually interrupt the flow.*
> >
> > We agree and we will fix this in the revised version.
> >
> > 16)  *Equation (11) should be rearranged as it is too long.*
> >
> > We agree and we will fix this in the revised version.
> >
> > 17)  *The name RRMSE is weird, as MSE typically represents mean squared error, while the actual definition is not the $\ell_2$ norm squared.*
> >
> > By RRMSE, we meant the relative root mean squared error given by $\frac{\|\boldsymbol{\tilde{\beta}}-\boldsymbol{\beta^*}\|_2}{\|\boldsymbol{\beta^*}\|_2}$. In our manuscript, we called it `relative reconstruction error', which is confusing. We will fix this in the revised version and refer to it as RRMSE all through the paper. The RRMSE is useful as it is a fraction from 0 to 1 and independent of the signal magnitude unlike the MSE.

---

> > > ### Author Response · Authors · 2025-01-09
> > > **Request to reviewer regarding any further changes to be made**
> > >
> > > We have submitted detailed responses to the queries raised by the reviewer. We are preparing our revision. Based on our responses, we request the reviewer to let us know whether any further changes are expected.

---

### Review · Reviewer_s3kr · 2024-12-13

**Summary Of Contributions:**

This paper addresses the problem of unlabelled compressive sensing, which involves estimating an unknown sparse vector from underdetermined linear measurements where the correspondence between some measurements and the rows of the sensing matrix is unknown. Two optimization approaches are proposed to estimate the sparse vector: the first adapts the LASSO problem to incorporate known correspondences, while the second employs a hard-thresholding pursuit method. Theoretical guarantees are provided for both algorithms. Through simulations, the proposed methods demonstrate superior performance compared to previous algorithms and highlight the advantage of leveraging known correspondences.

**Audience:**

Yes

**Claims And Evidence:**

Yes

**Requested Changes:**

I suggest that the authors provide a more comprehensive set of experiments, including comparisons with state-of-the-art compressive sensing algorithms and evaluations on more practical applications. Additionally, the performance of the proposed methods should be tested across different dimensionalities and sparsity levels to better understand their behavior in various scenarios.

It would also be beneficial to include a computational comparison of the proposed algorithms against other methods to highlight their efficiency.

Finally, Figure 1 in the Supplementary Material is difficult to interpret. I recommend improving the caption for clarity or adding direct labels to each image for better readability.

**Strengths And Weaknesses:**

Strengths:

1- The paper proposes practical algorithms with theoretical guarantees for cases where some correspondences between measurements and the rows of the sensing matrix are known, which is critical in certain applications.

2- The paper includes a comprehensive literature review of related works and effectively positions its contributions within the existing body of knowledge.

Weaknesses:

1- The experiments are limited, as the effects of varying sparsity parameters and the dimensions of the sensing matrix have not been thoroughly evaluated. Additionally, the practical example provided is insufficiently detailed. More extensive results are needed to fully understand the behavior of the proposed algorithms across different dimensions, particularly in highly underdetermined cases.

2- While numerous powerful compressive sensing algorithms exist for this well-established problem, these algorithms are missing from the experimental comparisons. This omission makes it difficult to accurately assess the significance of the proposed methods.

3- The discussion of results is insufficient. For example, there is no explanation for why the performance of the L1-L1 algorithm is significantly worse than others.

4- The proposed algorithms involve several important hyperparameters, but no computationally efficient or robust method is provided for tuning them apart from standard cross-validation. Furthermore, the sensitivity of the algorithms to these hyperparameters has not been investigated.

---

> ### Author Response · Authors · 2024-12-13
> **Requesting for some more details regarding the comparisons with the state of the art techniques**
>
> We would like to sincerely thank the reviewer for their detailed review and for having read our manuscript carefully. We have understood most of the concerns raised by the reviewer, and will incorporate various changes to improve the clarity of the manuscript, and to also present a more detailed set of results. However, there is one comment, mentioned below, which we did not fully understand. We would be grateful if reviewer could provide us more details.
>
> The reviewer states:
>
> \textit{While numerous powerful compressive sensing algorithms exist for this well-established problem, these algorithms are missing from the experimental comparisons. This omission makes it difficult to accurately assess the significance of the proposed methods.}
>
> and
>
> \textit{I suggest that the authors provide a more comprehensive set of experiments, including comparisons with state-of-the-art compressive sensing algorithms ...}
>
> We would like to re-emphasize that our manuscript deals with \emph{unlabeled} compressive sensing, i.e. we are specifically handling the case where some of the measurements are permuted. There truly exist many compressive sensing algorithms, but they do not deal with compressive \emph{unlabeled} sensing. The only one we could find after extensive literature search is the following one:
>
> [RA] Liangzu Peng, Boshi Wang, and Manolis Tsakiris. Homomorphic sensing: Sparsity and noise. In Int. Conf.
> Machine Learning, 2021
>
> We have therefore experimentally compared our technique with that in [RA] and pointed out various ways in which our work differs from that in [RA].  We'd be really grateful if the reviewer could tell us with which other algorithms for unlabeled compressive sensing they would like us to perform a comparison. We will do our best to perform a thorough comparison.
>
> Once again, we would like to thank the reviewer for their efforts in reviewing our manuscript.

---

> > ### Comment · Reviewer_s3kr · 2024-12-13
> >
> > I appreciate the authors' response and apologize for any lack of clarity in my previous comments.
> >
> > To clarify, would it be possible for the authors to consider using more efficient compressed sensing solvers than the L1 solver and HTP, adapting them to the case of unlabeled CS with some known correspondences? There are significantly more efficient solvers available for the standard CS problem, and I'm wondering if they could be similarly adapted.

---

> > > ### Author Response · Authors · 2024-12-13
> > >
> > > We thank the reviewer for their reply. The reviewer states:
> > >
> > > “To clarify, would it be possible for the authors to consider using more efficient compressed sensing solvers than the L1 solver and HTP, adapting them to the case of unlabeled CS with some known correspondences? There are significantly more efficient solvers available for the standard CS problem, and I'm wondering if they could be similarly adapted.”
> > >
> > > The LASSO, which we have adapted for unlabeled sensing, is an L1-based solver, and it is proved to be minimax optimal (see Theorem 11.1, Example 11.1 and comments after Equation 11.15  of the book “Statistical Learning with Sparsity: The LASSO and Generalizations” by Hastie, Tibshirani and Wainwright). Since the LASSO is minimax optimal, it is not possible to substantially improve on this estimator.
> > >
> > > Yes, learning-based methods for compressed sensing exist (if that is what the reviewer had in mind). But it is very hard to prove any theoretical performance bounds on learning-based estimators, whereas theoretical performance bounds are a very major contribution of our manuscript. Moreover, permutations can occur at any random subset of measurement indices, and hence it is very hard to use learning-based algorithms to correct for permutation error without a prohibitively large set (exponential in p, for p-dimensional vectors) of training samples.
> > >
> > > Likewise, we modified the HTP algorithm taking the method from “Foucart, S. Hard thresholding pursuit: An algorithm for compressive sensing. SIAM Journal on Numerical Analysis, 49(6):2543–2563, 2011.” as a baseline. In principle, we could have modified other greedy pursuit algorithms such as OMP (orthogonal matching pursuit) or CoSamp for the unlabeled sensing problem. But such modifications would not add any substantially newer ideas than what we have already presented in terms of performance bounds on the modified HTP.
> > >
> > > If the reviewer can mention some specific algorithm(s), with which they would like our method to be compared, we’ll be happy to do so.

---

> ### Comment · Reviewer_s3kr · 2025-02-09
>
> Thank you for preparing the revision. My concerns have been appropriately addressed.
>
> Meanwhile, I have a few minor comments:
>
> - In Figure 1, please use different markers in addition to (or instead of) color encoding to improve readability. Also, in Figure 1.e, consider using different line widths or styles, as some plots appear completely overlapped and unreadable.
>
> - In Figure 4, is the caption ordering (Row 1 - Row 3 - Row 2) intentional, or is it a mistake?
>
> - Please ensure consistency in naming the algorithms. For example, in the text, you use AR-LASSO, while in the title of Section 3.1, it is written as Ar-Lasso.
>
> - Please check the manuscript for typographical errors. For instance, on page 13: "The ℓ1 fidelity term term ...".

---

### Review · Reviewer_HVAM · 2024-12-15

**Summary Of Contributions:**

The paper considers the compressed sensing problem under the assumption that labels are partially shuffled. The problem has been extensively studied in both overparametrized settings and underparametrized settings. The innovation here is considering a middle ground where some labels are aligned. The paper's main result is the sample complexity of the proposed hard thresholding algorithm.

**Audience:**

Yes

**Broader Impact Concerns:**

No concern.

**Claims And Evidence:**

Yes

**Requested Changes:**

In addition to the changes requested in the previous section:

* In the introduction, please clarify the contribution of the Suresh et al. 2024 paper and how this paper improves them.
* Focus on three key applications of unlabeled CS and demonstrate performance against benchmarks stated in the introduction
* In the contribution section, summarize the contribution of the proof techniques used.

**Strengths And Weaknesses:**

Strength
* Clear problem setting and problem statement

* Earlier literature on unlabeled compressed sensing has been reviewed

Weakness
* The organization of the paper can be improved. It took me a lot of time to get what "n" is the third contribution item on page 3  before reading on and finding the definition of n later in that section.

* I am slightly confused by the bound in contribution item #3 on page 3. N is lower bounded by C log(n) and n = N -m. Is that correct. It would be great to have a table of notation and state which veriable is constant and which variable is growing with what rate.

* The paper talks about image registration as an application but needs to provide complete empirical evidence in the main text. I suggest bringing those results from the appendix to the main text and expanding on it.

* Figure 2 seems to miss the left panel and is incomplete

* Overall, the contribution of the paper is limited given the body of the work done in the area of shuffled regression and unlabelled compressed sensing

---

> ### Author Response · Authors · 2025-01-06
> **Response to reviewer HVAM: Part 1**
>
> We thank the reviewer for their detailed comments and are responding below.
>
> 1) *The organization of the paper can be improved. It took me a lot of time to get what $n$ is the third contribution item on page 3 before reading on and finding the definition of $n$ later in that section.*
>
> We agree with the reviewer, and we had missed precisely defining $n$ to be equal to $N-m$ (the number of measurements which could have a permutation in them) in the first place where $n$ was used. We will include a symbol table to make the revised paper easier to read.
>
> 2) *I am slightly confused by the bound in contribution item 3 on page 3. $N$ is lower bounded by $C \log n$ and $n = N -m$. Is that correct? It would be great to have a table of notation and state which variable is constant and which variable is growing with what rate.*
>
> We agree with the reviewer that a table of notations is required as part of the main paper. We will include it in the revised version.
> There is no bound of the form $N \geq C \log n$. We assume that the reviewer is referring to the bound $N \geq C(k\log{(p/3k)} + s\log{(n/3s)})$ where $N$ is the total number of measurements, $m$ is the number of measurements with known correspondences and $n = N-m$. Yes, this lower bound on $N$ (for the theorems in the paper to hold) is correct. In unlabeled compressed sensing, there are two unknown vectors: the signal vector $\boldsymbol{\beta} \in \mathbb{R}^p$ and the permutation error vector $\boldsymbol{e} \in \mathbb{R}^n$. The lower bound $N \geq C(k\log{(p/3k)} + s\log{(n/3s)})$ is a sufficient condition for an accurate recovery of $\boldsymbol{\beta}^*$ via A-HTP (see Theorem 2 and Lemma 3 in the manuscript). The detailed proof of Lemma 3 is given in Lemma S4 in the supplementary material.
>
> 3) *The paper talks about image registration as an application but needs to provide complete empirical evidence in the main text. I suggest bringing those results from the appendix to the main text and expanding on it.*
>
> Yes, we agree to include it in the main paper in the revised version, and improving upon the explanation (see response to reviewer s3kr titled  `detailed explanation of the image alignment experiment').
>
> 4) *Figure 2 seems to miss the left panel and is incomplete.*
>
> Actually, this figure is complete, but we will align the figure to be in the center and we will rewrite its caption as follows: ``Distribution of the relative root mean squared error (RRMSE) values obtained using $\ell_1$-HTP and $\ell_2$-HTP for different instances of permutation matrix and measurement noise for $N = 80$ (left) and $N = 90$ (right) measurements. Utilizing the prior known correspondences (as done by $\ell_2$-HTP) results in a much lower standard deviation of the errors.''
>
> 5) *Overall, the contribution of the paper is limited given the body of the work done in the area of shuffled regression and unlabelled compressed sensing.*
>
> The presented techniques are simple, but we wish to emphasize that ours is the first piece of work to present a theoretical analysis of unlabeled compressed sensing for any tractable estimator. The literature on unlabeled compressed sensing is very limited (only one paper on this topic exists to the best of our knowledge). We have new concepts such as the GEREC and the SS-RIP which are useful in the theoretical derivations for AR-LASSO and AR-HTP respectively.
>
> 6) *In the introduction, please clarify the contribution of the Suresh et al. 2024 paper and how this paper improves them.*
>
> We assume the reviewer is referring to the paper `Sresth et al 2024'. If so, then their work considers the problem of unlabeled sensing in the case where the number of measurements $N$ exceeds the signal dimension $p$. In contrast to that, the present manuscript looks at the unlabeled compressive sensing problem where $N \ll p$. Hence, the theoretical and algorithmic treatment is totally different in the two papers, and the two papers are working on two completely different problems.
>
> 7) *Focus on three key applications of unlabeled CS and demonstrate performance against benchmarks stated in the introduction.*
>
> We have performed comparisons with an existing unlabeled compressed sensing technique which uses the $\ell_1$-HTP algorithm. We have also explored an application in image registration. We consider a deeper exploration of a larger number of applications to be beyond the scope of this work, which focuses primarily on theoretical bounds for tractable estimators.

---

> > ### Author Response · Authors · 2025-01-06
> > **Response to reviewer HVAM: Part 2**
> >
> > 8) *In the contribution section, summarize the contribution of the proof techniques used.*
> >
> > We agree with the reviewer that this part should be better written. We will add these details in the revised version. The key contributions in the proof techniques used include: (1) the proposition of generalised, extended, restricted eigenvalue condition (GEREC) and its use within the AR-LASSO bounds; (2) concept of structured-sparsity, restricted isometric property (SS-RIP) and the use of the specific form of structured sparsity in the $(p+n)$-dimensional vector $\boldsymbol{x}^* := \begin{bmatrix}
> > {\boldsymbol{\beta}^*}^T &
> > {\boldsymbol{z}^*}^T
> > \end{bmatrix}^T$ being recovered -- namely that the sub-vector corresponding to $\boldsymbol{\beta} \in \mathbb{R}^p$ contains $k$ non-zero elements, and the sub-vector corresponding to $\boldsymbol{e} \in \mathbb{R}^n$ contains $s$ non-zero elements. The use of such a structure to obtain a requirement on number of measurements for an accurate recovery of $\boldsymbol{x}^*$ via A-HTP is novel, and is not found in classic papers on model based compressive sensing such as \cite{baraniuk2010model}.

---

> > > ### Comment · Reviewer_HVAM · 2025-01-06
> > > **Thanks for the response**
> > >
> > > My comments are addressed. Please make sure to update the manuscript accordingly. I am updating my score as well.

---

> > > > ### Author Response · Authors · 2025-02-20
> > > > **Request for recommendation on the revised manuscript**
> > > >
> > > > Dear reviewer HVAM,
> > > >
> > > > Once again, thank you for your detailed review. We have responded to your comments (and those by the other two reviewers) and also submitted our revision with changes marked in blue font. We have also included detailed point for point responses to concerns raised by all three reviewers as a separate document, now part of the supplemental material. We request you to submit your recommendation so that the editor may arrive at a decision.
> > > >
> > > > Sincerely,
> > > > Authors

---

### Comment · Action_Editor_3d9S · 2025-01-29
**Revision**

Dear authors and reviewers,

I've noticed that while all reviewers are mostly positive, they've all requested changes, and the authors have promised changes but didn't actually submit a revision.  I think think there is a misunderstanding on how the review process for TMLR works.  I apologize for not following the discussions more closely, because I should have pointed this out earlier.

With TMLR, the authors can upload a new manuscript at any point! There is no "second round". So I am requesting that
- the authors **upload their revision within the next week**
- after that, the **reviewers have one week** to look at it.
If there are no issues raised by the reviewers after that week then I'll go ahead with the decision.

Again, apologies for not pointing this out earlier. I know this will delay the paper, but I think it's worth it.

Best,
Stephen (action editor)

---

> ### Comment · Action_Editor_3d9S · 2025-02-07
> **Planning to reject if no revision uploaded**
>
> Dear authors,
>
> To reiterate what I wrote a week ago, all the reviewer comments were contingent on a revision mentioned in your response.  I do not see that revision uploaded yet.  From their reviews (not visible to you), "The rebuttal promises many changes, and I look forward to reading them in the revision" and (from a different reviewer) "I believe that the paper would be in an appropriate shape after applying the required changes."
>
> Please upload your revision within the next two days, after which I'll give the reviewers a week to look over the changes.
>
> If we can't make this timeframe then I think we should reject and have the paper be resubmitted as a new submission.
>
> Best,
> Stephen

---

> > ### Comment · Action_Editor_3d9S · 2025-02-08
> > **Reviewers, please look at revision**
> >
> > Thanks to the authors for uploading a revision.
> >
> > Reviewers, please take a look at the revision and see if it is what you were asking for.  I'd really like to finish this round within the week if possible, so thanks in advance for being quick.
> >
> > Best,
> >
> > Stephen

---

> > > ### Comment · Reviewer_9Gpt · 2025-02-09
> > > **I checked the revision**
> > >
> > > Dear editor, I have taken a look at the revision (the authors' response is in the supplementary pdf and the added text of the main manuscript in blue). It has addressed almost all of my points.

---

> > ### Author Response · Authors · 2025-02-08
> > **Revision submitted**
> >
> > Dear editor,
> >
> > We have now submitted our revision. We have responded to all reviewer comments in a separate document called responses_reviewers.pdf as part of the supplemental material. All major changes made to the main paper or the supplemental material are highlighted in blue font.
> >
> > We sincerely thank the editor and all three reviewers for their detailed comments. We believe that addressing their concerns has improved the presentation of our paper and helped clarify many important details.
> >
> > Thank you,
> > Authors

---

> ### Comment · Reviewer_s3kr · 2025-02-09
> **Confirmation of the revision + minor comments**
>
> Dear Editor,
>
> I have read the revised version of the manuscript and confirm that my concerns have been addressed. I have provided some minor comments.

---

### Decision · Action_Editor_3d9S · 2025-02-25

**Recommendation:** Accept with minor revision

**Comment:**

Apologies for the unusual reviewing cycle.  The authors have now provided a revision, and 2 of 3 reviewers are happy with that (the 3rd reviewer did not respond). The authors did a good job engaging in the rebuttal and making changes. I think the review process has done its job and improved the paper.

Thus I'm happy to recommend acceptance. However, reviewer `s3kr` mentions a few new minor comments (see the bottom of their original thread) that should be addressed before the final camera-ready version.

**Audience:**

To quote from a reviewer, "The paper considers the compressed sensing problem under the assumption that labels are partially shuffled. The problem has been extensively studied in both overparametrized settings and underparametrized settings. The innovation here is considering a middle ground where some labels are aligned. The paper's main result is the sample complexity of the proposed hard thresholding algorithm"

Compressed sensing is by now a large topic of wide interest. However, because of that, there are fewer fundamental results left to be shown. This particular papers addresses a very special subcase (**some** labels are known).  I would describe this as interesting to some readers, though not a "big" topic that would interest all readers.  Overall, I think it's certainly true that "at least some individuals in TMLR's audience be interested in knowing the findings of this paper".

**Claims And Evidence:**

The paper provides theory, experiments and applications to the problem of unlabeled compressed sensing.  There were no concerns raised over theory, and numerical experiments were largely satisfactory too (with some discussion during the rebuttal phase of other baselines). Overall, reviewers did not raise objections about the correctness of the results.